# A Language Agent for Autonomous Driving

**Jiageng Mao**[1]* **Junjie Ye**[1]* **Yuxi Qian**[1] **Marco Pavone**[2,3] **Yue Wang**[1,3]
[1]University of Southern California  [2]Stanford University  [3]NVIDIA
{jiagengm, yejunjie, yuxiqian, yue.w}@usc.edu, pavone@stanford.edu

## Abstract

Human-level driving is an ultimate goal of autonomous driving. Conventional approaches formulate autonomous driving as a perception-prediction-planning framework, yet their systems do not capitalize on the inherent reasoning ability and experiential knowledge of humans. In this paper, we propose a fundamental paradigm shift from current pipelines, exploiting Large Language Models (LLMs) as a cognitive agent to integrate human-like intelligence into autonomous driving systems. Our system, termed Agent-Driver, transforms the traditional autonomous driving pipeline by introducing a versatile tool library accessible via function calls, a cognitive memory of common sense and experiential knowledge for decision-making, and a reasoning engine capable of chain-of-thought reasoning, task planning, motion planning, and self-reflection. Powered by LLMs, our Agent-Driver is endowed with intuitive common sense and robust reasoning capabilities, thus enabling a more nuanced, human-like approach to autonomous driving. We evaluate our system on both open-loop and close-loop driving challenges, and extensive experiments substantiate that our Agent-Driver significantly outperforms the state-of-the-art driving methods by a large margin (more than 30% on the nuScenes dataset). Our approach also demonstrates superior interpretability and few-shot learning ability to these methods. Please visit our webpage for more details.

## 1 Introduction

Imagine a car navigating a quiet suburban neighborhood. Suddenly, a ball bounces onto the road. A human driver, leveraging extensive experiential knowledge, would not only perceive the immediate presence of the ball, but also instinctively anticipate the possibility of a chasing child and consequently decide to decelerate. In contrast, an autonomous vehicle, devoid of such reasoning and experiential anticipation, might continue driving until sensors detect the child, only allowing for a narrower margin of safety. The importance of human

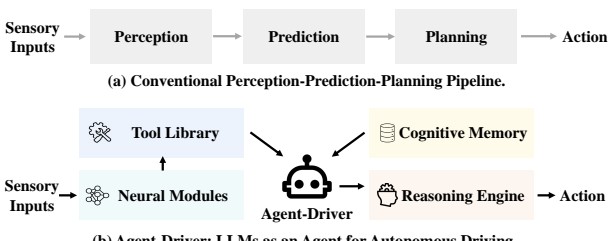

(a) Conventional Perception-Prediction-Planning Pipeline.

(b) Agent-Driver: LLMs as an Agent for Autonomous Driving.

Figure 1: **Comparison between (a) the conventional driving system and (b) the proposed Agent-Driver.** Our approach transforms the conventional perception-prediction-planning framework by introducing LLMs as an agent for autonomous driving.

prior knowledge in driving systems becomes clear: driving is not merely about reacting to the visible, but also to the conceivable scenarios where the system needs to reason and respond even in their absence.

To integrate human prior knowledge into autonomous driving systems, previous approaches (Sadat et al., 2020; Casas et al., 2021; Hu et al., 2022; 2023; Jiang et al., 2023)

---

*indicates equal contribution.

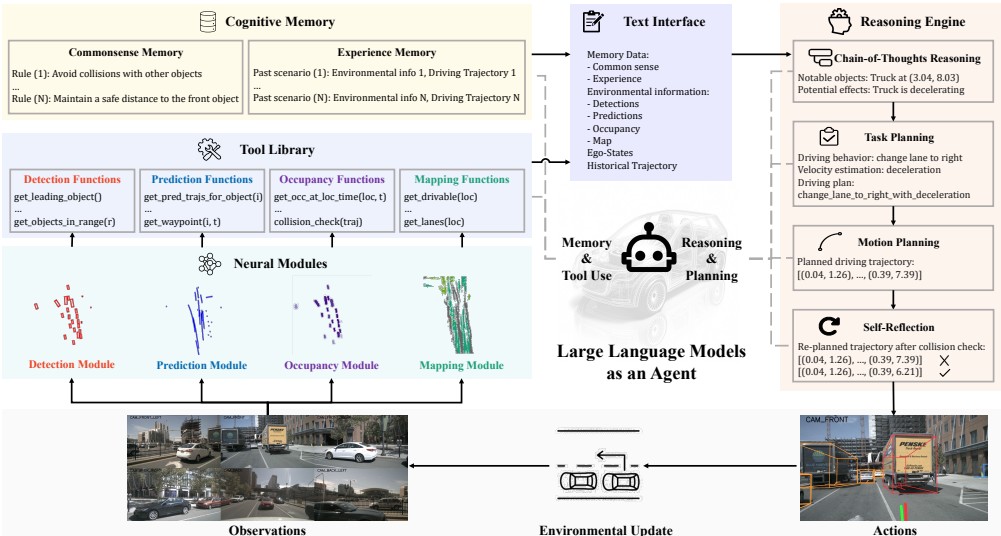

Figure 2: **Architecture of Agent-Driver.** Our system dynamically collects necessary environmental information from the output of neural modules via the **tool library**. The collected information is further utilized to query the **cognitive memory**. Consequently, the **reasoning engine** takes collected environmental information and retrieved memory data as input, and traceably derives a safe and comfortable trajectory for driving through chain-of-thought reasoning, task planning, motion planning, and self-reflection.

deconstruct the human driving process into three systematic steps following Figure 1 (a). *Perception*: they interpret the human perceptual process as object detection (Mao et al., 2023b) or occupancy estimation (Peng et al., 2020). *Prediction*: they abstract human drivers' foresight of upcoming scenarios as the prediction of future object motions (Casas et al., 2018). *Planning*: they emulate the human decision-making process by planning a collision-free trajectory, either using hand-crafted rules (Treiber et al., 2000) or by learning from data (Zeng et al., 2019). Despite its efficacy, this *perception-prediction-planning* framework overly simplifies the human decision-making process and cannot fully model the complexity of driving. For instance, perception modules in these methods are notably redundant, necessitating the detection of all objects in a vast perception range, whereas human drivers can maintain safety by only attending to a few key objects. Moreover, prediction and planning are designed for collision avoidance with detected objects. Nevertheless, they lack deeper reasoning ability inherent to humans, *e.g.* deducing the connection between a visible ball and a potentially unseen child. Furthermore, it remains challenging to incorporate long-term driving experiences and common sense into existing autonomous driving systems.

In addressing these challenges, we found the major obstacle of integrating human priors into autonomous driving lies in the incompatibility of human knowledge and neural-network-based driving systems. Human knowledge is inherently encoded and utilized as language representations, and their reasoning process can also be interpreted by language. However, conventional driving systems rely on deep neural networks that are designed to process numerical data inputs, such as sensory signals, bounding boxes, and trajectories. The discrepancy between language and numerical representations poses a significant challenge to incorporating human experiential knowledge and reasoning capability into existing driving systems, thereby widening the chasm from genuine human driving performance.

Taking a step towards more human-like autonomous driving, we propose Agent-Driver, a cognitive agent empowered by Large Language Models (LLMs). The fundamental insight of our approach lies in the utilization of natural language as a unified interface, seamlessly bridging language-based human knowledge and reasoning ability with neural-network-based systems. Our approach fundamentally transforms the conventional perception-prediction-planning framework by leveraging LLMs as an interactive scheduler among system components. As depicted in Figure 1 (b), on top of the LLMs, we introduce: 1) a versatile tool library that interfaces with neural modules via dynamic function calls,

streamlining perception with less redundancy, 2) a configurable cognitive memory that explicitly stores common sense and driving experiences, infusing the system with human experiential knowledge, and 3) a reasoning engine that processes perception results and memory data to emulate human-like decision-making. Specifically, the reasoning engine performs chain-of-thought reasoning to recognize key objects and events, task planning to derive a high-level driving plan, motion planning to generate a driving trajectory, and self-reflection to ensure the safety of the planned trajectory. These components, coordinated by LLMs, culminate in an anthropomorphic driving process. To conclude, we summarize our contributions as follows:

- We present Agent-Driver, an LLM-powered agent that revolutionizes the traditional perception-prediction-planning framework, establishing a powerful yet flexible paradigm for human-like autonomous driving.
- Agent-Driver integrates a tool library for dynamic perception and prediction, a cognitive memory for human knowledge, and a reasoning engine that emulates human decision-making, all orchestrated by LLMs to enable a more anthropomorphic autonomous driving process.
- Agent-Driver significantly outperforms the state-of-the-art autonomous driving systems by a large margin, with over 30% collision improvements in motion planning. Our approach also demonstrates strong few-shot learning ability and interpretability on the nuScenes benchmark.
- We provide ablation studies to dissect the proposed architecture and understand the efficacy of each module, to facilitate future research in this direction.

## 2 Related Works

**Perception-Prediction-Planning in Driving Systems.** Modern autonomous driving systems rely on a perception-prediction-planning paradigm to make driving decisions based on sensory inputs. Perception modules aim to recognize and localize objects in a driving scene, typically in a format of object detection (Mao et al., 2021b; Wang et al., 2022; Mao et al., 2021a; 2023b) or object occupancy prediction (Peng et al., 2020; Tong et al., 2023). Prediction modules aim to estimate the future motions of objects, normally represented as predicted trajectories (Casas et al., 2018; Ivanovic & Pavone, 2019; Shi et al., 2022) or occupancy flows (Agro et al., 2023; Casas et al., 2021). Planning modules aim to derive a safe and comfortable trajectory, using rules (Urmson et al., 2008; Fan et al., 2018; Sauer et al., 2018; Chen et al., 2015; Bacha et al., 2008; Leonard et al., 2008; Thrun et al., 2006; Treiber et al., 2000) or learning from human driving trajectories (Dauner et al., 2023; Chitta et al., 2022; Mao et al., 2023a). These three modules are generally performed sequentially, either trained separately or in an end-to-end manner (Liang et al., 2020; Sadat et al., 2020; Casas et al., 2021; Hu et al., 2022; 2023). This perception-prediction-planning framework overly simplifies the human driving process and cannot effectively incorporate human priors such as common sense and past driving experiences. By contrast, our Agent-Driver transforms the conventional perception-prediction-planning framework by introducing LLMs as an agent to bring human-like intelligence into the autonomous driving system.

**LLMs in Autonomous Driving.** Trained on Internet-scale data, LLMs (Brown et al., 2020; OpenAI, 2023; Touvron et al., 2023a;b) have demonstrated remarkable capabilities in commonsense reasoning and natural language understanding. How to leverage the power of LLMs to tackle the problem of autonomous driving remains an open challenge. GPT-Driver (Mao et al., 2023a) handled the planning problem in autonomous driving by reformulating motion planning as a language modeling problem and introducing fine-tuned LLMs as a motion planner. DriveGPT4 (Xu et al., 2023) proposed an end-to-end driving approach that leverages Vision-Language Models to directly map sensory inputs to actions. DiLu (Wen et al., 2023) introduced a knowledge-driven approach with Large Language Models. These methods mainly focus on an individual component in conventional driving systems, e.g. question-answering (Xu et al., 2023), planning (Mao et al., 2023a), or control (Sha et al., 2023). Some approaches (Fu et al., 2023; Wen et al., 2023) are implemented and evaluated in simple simulated driving environments. By contrast, Agent-Driver presents a systematic approach

that leverages LLMs as an agent to schedule the whole driving system, leading to a strong performance on the real-world driving benchmark.

## 3  Agent-Driver

In this section, we present Agent-Driver, an LLM-based intelligent agent for autonomous driving. We first introduce the overall architecture of our Agent-Driver in Section 3.1. Then, we introduce the three key components of our method: tool library (Section 3.2), cognitive memory (Section 3.3), and reasoning engine (Section 3.4).

### 3.1  Overall Architecture

Conventional perception-prediction-planning pipelines leverage a series of neural networks as basic modules for different tasks. However, these neural-network-based systems lack direct compatibility with human prior knowledge, constraining their potential for leveraging such priors to enhance driving performance. To handle this challenge, we propose a novel framework that leverages text representations as a unified interface to connect neural networks and human knowledge. The overall architecture of Agent-Driver is shown in Figure 2. Our approach takes sensory data as input and introduces neural modules for processing these sensory data and extracting environmental information about detection, prediction, occupancy, and map. On top of the neural modules, we propose a **tool library** where a set of functions are designed to further abstract the neural outputs and return text-based messages. For each driving scenario, an LLM selectively activates the required neural modules by invoking specific functions from the tool library, ensuring the collection of necessary environmental information with less redundancy. Upon gathering the necessary environmental information, the LLM leverages this data as a query to search in a **cognitive memory** for pertinent traffic regulations and the most similar past driving experience. Finally, the retrieved traffic rules and driving experience, together with the formerly collected environmental information, are utilized as inputs to an LLM-based **reasoning engine**. The reasoning engine performs multi-round reasoning based on the inputs and eventually devises a safe and comfortable trajectory for driving. Our Agent-Driver architecture harnesses dynamic perception and prediction capability brought by the tool library, human knowledge from the cognitive memory, and the strong decision-making ability of the reasoning engine. This synergistic integration results in a more human-like driving system with enhanced decision-making capability.

### 3.2  Tool Library

The profound challenge of incorporating human knowledge into neural-network-based driving systems is reconciling the incompatibility between text-based human priors and the numerical representations from neural networks. While prior works (Kuo et al., 2022) have attempted to translate text-based priors into semantic features or regularization terms for integration with neural modules, their performances are still constrained by the inherent cross-modal discrepancy. By contrast, we leverage text as a unified interface to connect neural modules and propose a tool library built upon the neural modules to dynamically collect text-based environmental information.

The cornerstones of the tool library are four neural modules, *i.e.*, detection, prediction, occupancy, and map modules, which process sensory data from observations and generate detected bounding boxes, future trajectories, occupancy grids, and maps respectively. The neural modules cover various tasks in perception and prediction and extract environmental information from observations. However, this information is largely redundant, with a significant portion insignificant to decision-making. To dynamically extract necessary information from the neural module outputs, we propose a tool library—where a set of functions are designed—to summarize the neural outputs into text-based messages, and the information collection process can be established by dynamic function calls. An illustration of this process is shown in Figure 3.

**Functions.** We devised various functions for detection, prediction, occupancy, and mapping, in order to extract useful information from the neural module's outputs respectively. Our

tool library contains more than 20 functions covering diverse usages. Here are some examples. For detection, `get_leading_object` returns a text description of the object in front of the ego-vehicle on the same lane. For prediction, `get_pred_trajs_for_object` returns a text-based predicted future trajectory for a specified object. For occupancy, `get_occ_at_loc_time` returns the probability that a specific location is occupied by other objects at a given timestep. For map, `get_lanes` returns the information of the left and right lanes to the ego-vehicle, and `get_shoulders` returns the information of the left and right road shoulders to the ego-vehicle. Detailed descriptions of all functions are in Appendix A.1.

**Tool Use.** With the functions in the tool library, an LLM is instructed to collect necessary environmental information through dynamic function calls. Specifically, the LLM is first provided with initial information such as the current state for its subsequent decision-making. Then, the LLM will be asked whether it is necessary to activate a specific neural module, *i.e.* detection, prediction, occupancy, and map. If the LLM decides to activate a neural module, the functions related to this module will be provided to the LLM, and the LLM chooses to call one or some of these functions to collect the desired information. Through multiple rounds of conversations, the LLM eventually collects all necessary information about the current environment. Compared to directly utilizing the outputs of the neural modules, our approach reduces the redundancy in current systems by leveraging the reasoning power of the LLM to determine what environmental information is of real importance to the decision-making process. Furthermore, the neural modules are only activated when the LLM decides to call the relevant functions, which brings flexibility to the system.

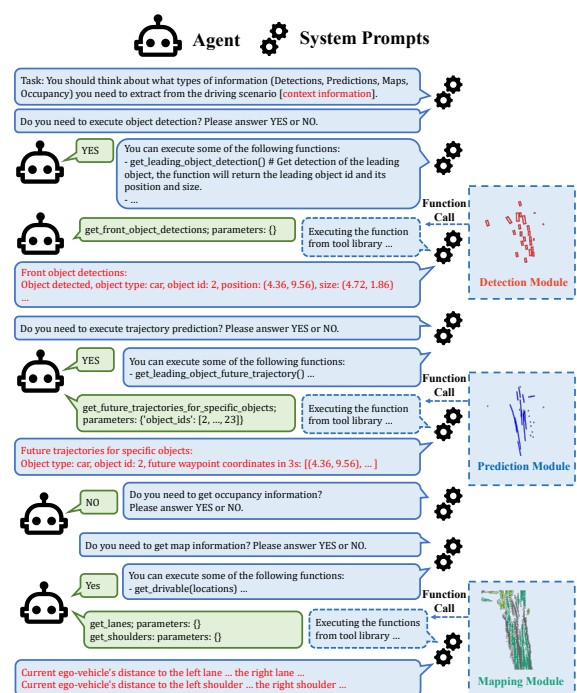

Figure 3: **Illustration of function calls in the tool library.** Agent-Driver can effectively collect necessary environmental information from neural modules through dynamic function calls.

A detailed example of LLM leveraging tool functions for environment perceiving is shown in Appendix A.2.

## 3.3 Cognitive Memory

Human drivers navigate using their common sense, such as adherence to local traffic regulations, and draw upon driving experiences in similar situations. However, it is non-trivial to adapt this ability to the conventional perception-prediction-planning framework. By contrast, our approach tackles this problem through interactions with a cognitive memory. Specifically, the cognitive memory stores text-based common sense and driving experiences. For every scenario, we utilize the collected environmental information as a query to search in the cognitive memory for similar past experiences to assist decision-making. The cognitive memory contains two sub-memories: commonsense memory and experience memory.

**Commonsense Memory.** The commonsense memory encapsulates the essential knowledge a driver typically needs for driving safely on the road, such as traffic regulations and knowledge about risky behaviors. It is worth noting that the commonsense memory is purely text-based and fully configurable, that is, users can customize their own commonsense memory for different driving conditions by simply writing different types of knowledge into the memory.

**Experience Memory.** The experience memory contains a series of past driving scenarios, where each scenario is composed of the environmental information and the subsequent driving decision at that time. By retrieving the most similar experiences and referencing their driving decisions, our system enhances its capacity for making more informed and resilient driving decisions.

**Memory Search.** As exhibited in Figure 4, we present an innovative two-stage search algorithm to effectively search for the most similar past driving scenario in the experience memory. The first stage of our algorithm is inspired by vector databases (Lewis et al., 2020; Wang et al., 2021), where we encode the input query and each record in the memory into embeddings and then retrieve the top-K similar records via K-nearest neighbors (K-NN) search in the embedding space. Since the driving scenarios are quite diverse, the embedding-based search is inherently limited by the encoding methods employed, resulting in insufficient generalization capabilities. To overcome this challenge, the second stage incorporates an LLM-based fuzzy search, where the LLM is tasked to rank these records according to their relevance to the query. This ranking is based on the implicit similarity assessment by the LLM, leveraging its capabilities in generalization and reasoning.

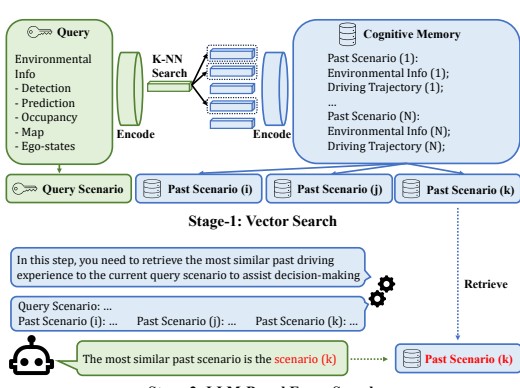

Figure 4: **Illustration of memory search.** The proposed two-stage search algorithm effectively retrieves the most similar driving experience and facilitates the subsequent decision-making process.

The cognitive memory equips our system with human knowledge and past driving experiences. The retrieved most similar experiences, together with commonsense and environmental information, collectively form the inputs to the reasoning engine. Text as a unified interface aligns the environmental information with human knowledge, thereby enhancing our system's compatibility. Please refer to Appendix B for more details.

### 3.4 Reasoning Engine

Reasoning, a fundamental ability of humans, is critical to the decision-making process. Conventional methods directly plan a driving trajectory based on perception and prediction results, while they lack the reasoning ability inherent to human drivers, resulting in insufficient capability to handle complicated driving scenarios. Conversely, as shown in Figure 5, we propose a reasoning engine that effectively incorporates reasoning ability into the driving decision-making process. Given the environmental information and retrieved memory, our reasoning engine performs multi-round reasoning to plan a safe and comfortable driving trajectory. The proposed reasoning engine consists of four core components: chain-of-thought reasoning, task planning, motion planning, and self-reflection.

**Chain-of-Thought Reasoning.** Human drivers are able to identify the key objects and their potential effects on driving decisions, while this important capability is typically absent in conventional autonomous driving approaches. To embrace this reasoning ability in our system, we propose a novel chain-of-thought reasoning module, where we instruct an LLM to reason on the input environmental information and output a list of key objects and their potential effects in text. To guide this reasoning process, we instruct the LLM via in-context learning of a few human-annotated examples. We found this strategy successfully aligns the reasoning power of the LLM with the context of autonomous driving, leading to improved reasoning accuracy.

**Task Planning.** High-level driving plans provide essential guidance to low-level motion planning. Nevertheless, traditional methods directly perform motion planning without relying on this high-level guidance, leading to sub-optimal planning results. In our approach, we define high-level driving plans as a combination of discrete driving behaviors and velocity estimations. For instance, the combination of a driving behavior

change_lane_to_left and a velocity estimation deceleration results in a high-level driving plan change_lane_to_left_with_deceleration. We instruct an LLM via in-context learning to devise a high-level driving plan based on environmental information, memory data, and chain-of-thought reasoning results. The devised high-level driving plan characterizes the ego-vehicle's coarse locomotion and serves as a strong prior to guide the subsequent motion planning process.

**Motion Planning.** Motion planning aims to devise a safe and comfortable trajectory for driving, and each trajectory is represented as a sequence of waypoints. Following (Mao et al., 2023a), we re-formulate motion planning as a language modeling problem. Specifically, we leverage environmental information, memory data, reasoning results, and high-level driving plans collectively as inputs to an LLM, and we instruct the LLM to generate text-based driving trajectories by reasoning on the inputs. By fine-tuning with human driving trajectories, the LLM can generate trajectories that closely emulate human driving patterns. Finally, we transform the text-based trajectories back into real trajectories for system execution.

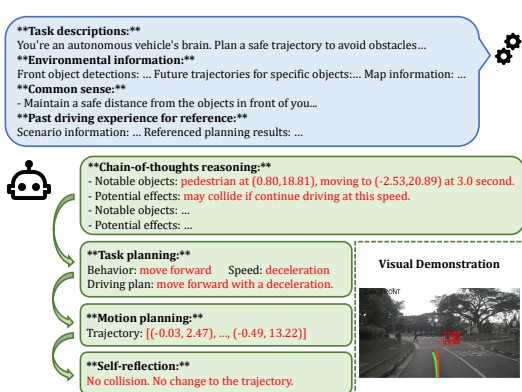

Figure 5: **Illustration of reasoning engine.** Agent-Driver makes driving decisions like human in a step-by-step procedure.

**Self-Reflection.** Self-reflection is a crucial ability in humans' decision-making process, aiming to re-assess the former decisions and adjust them accordingly. To model this capability in our system, we propose a collision check and optimization approach. Specifically, for a planned trajectory $\hat{\tau}$ from the motion planning module, the collision_check function in the tool library is first invoked to check its collision. If collision detected, we refine the trajectory $\hat{\tau}$ into a new trajectory $\tau^*$ by optimizing the cost function $\mathcal{C}$:

$$\tau^* = \min_\tau \mathcal{C}(\tau, \hat{\tau}) = \min_\tau \lambda_1 ||\tau - \hat{\tau}||_2 + \lambda_2 \mathcal{F}_{col}(\tau). \tag{1}$$

In this manner, the safety of the planned trajectory is greatly improved.

Our reasoning engine models the human decision-making process in driving as a step-by-step procedure involving reasoning, hierarchical planning, and self-reflection. Compared to prior works, our approach effectively emulates the human decision-making process, leading to enhanced decision-making capability and superior planning performance. More details of the reasoning engine can be found in Appendix C.

## 4 Experiments

In this section, we demonstrate the effectiveness, few-shot learning ability, and other characteristics of Agent-Driver through extensive experiments. First, we introduce the experimental settings in Section 4.1. Next, we evaluate the planning performance of our approach on both open-loop and closed-loop settings (Section 4.2). Subsequently, we investigate the few-shot learning ability (Section 4.3), interpretability (Section 4.4), compatibility (Section 4.5), and stability (Section 4.6) of Agent-Driver. Finally, we discuss the choices of in-context learning and fine-tuning in Section 4.7. More experiments can be found in Appendix D.

### 4.1 Experimental Setup

**Benchmarks.** For open-loop autonomous driving, we conduct experiments on the large-scale nuScenes dataset (Caesar et al., 2020). The nuScenes dataset is a real-world autonomous driving dataset that contains $1,000$ driving scenarios and approximately $34,000$ key frames encompassing a diverse range of locations and weather conditions. We follow the general practice and split the whole dataset into training and validation sets. We utilize the training set to train the neural modules and instruct the LLMs, and we utilize the validation set to

| | Method | L2 (m) ↓ | | | | Collision (%) ↓ | | | |
|---|---|---|---|---|---|---|---|---|---|
| | | 1s | 2s | 3s | Avg. | 1s | 2s | 3s | Avg. |
| *ST-P3 metrics* | ST-P3 (Hu et al., 2022) | 1.33 | 2.11 | 2.90 | 2.11 | 0.23 | 0.62 | 1.27 | 0.71 |
| | VAD (Jiang et al., 2023) | 0.17 | 0.34 | **0.60** | 0.37 | 0.07 | 0.10 | 0.24 | 0.14 |
| | GPT-Driver (Mao et al., 2023a) | 0.20 | 0.40 | 0.70 | 0.44 | 0.04 | 0.12 | 0.36 | 0.17 |
| | **Agent-Driver (ours)** | **0.16** | **0.34** | 0.61 | **0.37** | **0.02** | **0.07** | **0.18** | **0.09** |
| *UniAD metrics* | NMP (Zeng et al., 2019) | - | - | 2.31 | - | - | - | 1.92 | - |
| | SA-NMP (Zeng et al., 2019) | - | - | 2.05 | - | - | - | 1.59 | - |
| | FF (Hu et al., 2021) | 0.55 | 1.20 | 2.54 | 1.43 | 0.06 | 0.17 | 1.07 | 0.43 |
| | EO (Khurana et al., 2022) | 0.67 | 1.36 | 2.78 | 1.60 | 0.04 | 0.09 | 0.88 | 0.33 |
| | UniAD (Hu et al., 2023) | 0.48 | 0.96 | 1.65 | 1.03 | 0.05 | 0.17 | 0.71 | 0.31 |
| | GPT-Driver (Mao et al., 2023a) | 0.27 | 0.74 | 1.52 | 0.84 | 0.07 | 0.15 | 1.10 | 0.44 |
| | **Agent-Driver (ours)** | **0.22** | **0.65** | **1.34** | **0.74** | **0.02** | **0.13** | **0.48** | **0.21** |

Table 2: **Open-loop planning performance compared to the state-of-the-arts.** Agent-Driver significantly outperforms prior works in terms of L2 and collision rate. Our approach attains more than 30% performance gains in collisions compared to the state-of-the-art methods.

evaluate the performance of our approach, ensuring a fair comparison with prior works. Following prior works (Hu et al., 2022; 2023; Jiang et al., 2023), the L2 error and collision rate are reported to evaluate the planning performance. For the closed-loop autonomous driving, we adopt the Town05-Short benchmark (Prakash et al., 2021) powered by the CARLA simulator (Dosovitskiy et al., 2017) for evaluation. The Town05-Short benchmark consists of 10 challenging driving routes with 3 intersections each, including a high density of dynamic agents. We adopt the widely-used route completion and driving scores to evaluate the planning performance. Route Completion denotes the progress of driving on a route, and the Driving Score additionally takes comfort and safety into calculation. More details of the open-loop and closed-loop evaluation metrics can be found in Appendix D.1.

**Implementation Details.** We utilize gpt-3.5-turbo-0613 as the foundation LLM for different components in our system. For motion planning, we follow (Mao et al., 2023a) and fine-tune the LLM with human driving trajectories in the nuScenes training set for *one epoch*. For neural modules, we adopted the modules in (Hu et al., 2023). For closed-loop experiments, we leveraged the perception modules in LAV (Chen & Krähenbühl, 2022) and kept the other parts of our system the same. We also followed the same training setting and evaluation protocols in (Chen & Krähenbühl, 2022) for a fair comparison. More implementation details can be found in the appendix D.2.

## 4.2 Comparison with State-of-the-art Methods

**Open-Loop Results.** As shown in Table 2, Agent-Driver surpasses state-of-the-art methods in both metrics and decreases the collision rate of the second-best performance by a large margin. Specifically, under ST-P3 metrics, Agent-Driver realizes the lowest average L2 error and greatly reduces the average collision rates by **35.7%** compared to the second-best performance. Under UniAD metrics, Agent-Driver achieves an L2 error of 0.74 and a collision rate of 0.21%, which are **11.9%** and **32.3%** better than the second-best methods GPT-Driver (Mao et al., 2023a) and UniAD (Hu et al., 2023), respectively. The promising performance on the collision rate verifies the effectiveness of the reasoning ability of Agent-Driver, which considerably increases the safety of the proposed autonomous driving system.

**Closed-Loop Results.** To analyze the performance of our approach in closed-loop settings, we evaluate Agent-Driver against other state-of-the-art methods on the authoritative CARLA simulator. The results on the Town05-Short benchmark (Chitta et al., 2022) are shown in Table 1. Agent-Driver achieves the highest route completion, surpassing the second-best VAD (Jiang et al., 2023) by 4.1%. In terms of driving score, Agent-Driver also yields a performance of 57.33%, which is on par with the prior arts.

| Methods | Driving Score ↑ | Route Completion ↑ |
|---|---|---|
| CILRS (Codevilla et al., 2019) | 7.47 | 13.40 |
| LBC (Cui et al., 2021) | 30.97 | 55.01 |
| Transfuser (Prakash et al., 2021) | 54.52 | 78.41 |
| ST-P3 (Hu et al., 2022) | 55.14 | 86.74 |
| VAD (Jiang et al., 2023) | **64.29** | 87.26 |
| Agent-Driver (Ours) | 57.33 | **91.37** |

Table 1: **Closed-loop planning performance compared to the state-of-the-arts.** Agent-Driver yields the best route completion and an on-par driving score compared to prior arts.

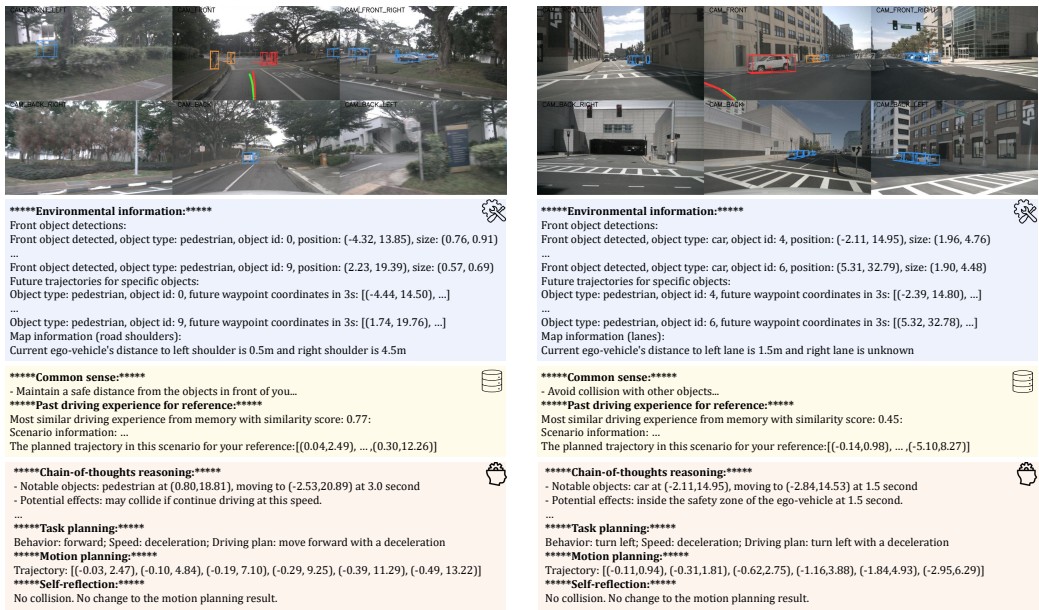

Figure 7: **Interpretability of Agent-Driver.** In the referenced images, planned trajectories of our system and human driving trajectories are in red and green respectively. Agent-Driver extracts meaningful objects (in yellow) from all detected objects (in blue) via the tool library. The reasoning engine further identifies notable objects (in red). Messages from the tool library, cognitive memory, and reasoning engine are recorded in colored text boxes. Every message is documented and our system is conducted in an interpretable and traceable way.

## 4.3 Few-shot Learning

To assess the generalization ability of motion planning in our approach, we conduct a few-shot learning experiment, where we keep other components the same and fine-tuned the core motion planning LLM with 0.1%, 1%, 10%, 50%, and 100% of the training data for *one epoch*. For comparison, we adopted the motion planner in UniAD (Hu et al., 2023) trained with 100% data as the baseline. The results are shown in Figure 6. Notably, with only 0.1% of the full training data, *i.e.*, 23 training samples, Agent-Driver realizes a promising performance. When exposed to 1% of training scenarios, the proposed method surpasses the baseline by a large margin, especially under the average collision rate. Furthermore, with increased training data, Agent-Driver stably achieves better motion planning performance.

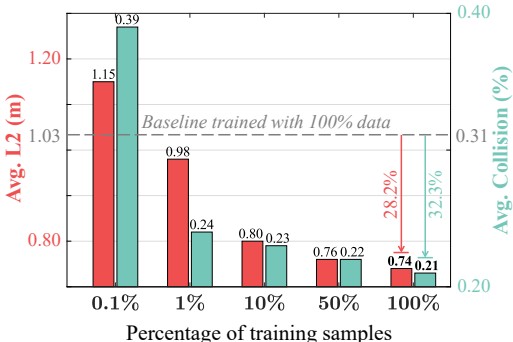

Figure 6: **Few-shot learning.** The motion planner in Agent-Driver fine-tuned with 1% data exceeds the state-of-the-art (Hu et al., 2023) trained on full data, verifying its few-shot learning ability.

## 4.4 Interpretability

Unlike conventional driving systems that rely on black-box neural networks to perform different tasks, the proposed Agent-Driver inherits favorable interpretability from LLMs. As shown in Figure 7, the output messages of LLMs from the tool library, cognitive memory, and reasoning engine are recorded during system execution. Hence the whole driving decision-making process is transparent and interpretable.

## 4.5 Compatibility with Different LLMs

We tried leveraging the Llama-2-7B (Tou-vron et al., 2023b), gpt-3.5-turbo-1106, and gpt-3.5-turbo-0613 models as the foundation LLMs in our system. Table 3 demonstrates that Agent-Driver powered by different LLMs can yield satisfactory performances, verifying the compatibility of our system with diverse LLM architectures.

| Method | L2 (m) ↓ | | | | Collision (%) ↓ | | | |
|---|---|---|---|---|---|---|---|---|
| | 1s | 2s | 3s | Avg. | 1s | 2s | 3s | Avg. |
| Llama-2-7B | 0.25 | 0.69 | 1.47 | 0.80 | 0.02 | 0.27 | 0.78 | 0.35 |
| gpt-3.5-turbo-1106 | 0.24 | 0.71 | 1.47 | 0.80 | 0.03 | 0.08 | 0.63 | 0.25 |
| gpt-3.5-turbo-0613 | 0.22 | 0.65 | 1.34 | 0.74 | 0.02 | 0.13 | 0.48 | 0.21 |

Table 3: **Compatibility to different LLMs.** Agent-Driver realizes satisfactory motion planning performance utilizing different types of LLMs as agents.

## 4.6 Stability

LLMs typically suffer from arbitrary predictions—they might produce invalid outputs (*e.g.*, hallucination or invalid formats)— which is detrimental to driving systems. To investigate this effect, we conducted a stability test of our Agent-Driver. Specifically, we used different amounts of training data to instruct the LLMs in our system, and we tested the number of invalid outputs during inference on the validation set, where we define "invalid" as waypoint outputs containing non-numerical values. As shown in Table 4, Agent-Driver exposed to only 1% of the training data sees *zero* invalid output during inference of 6,019 validation scenarios, suggesting that our system attains high output stability with proper instructions.

| Percentage of training samples | 0.10% | 1% | 10% | 50% | 100% |
|---|---|---|---|---|---|
| Number of invalid outputs | 2 | 0 | 0 | 0 | 0 |

Table 4: **Stability of Agent-Driver exposed to different amounts of training samples.** With only 1% training samples ($\sim$ 230 samples), Agent-Driver produces *zero* invalid output.

## 4.7 In-Context Learning vs. Fine-Tuning

Two prevalent strategies to instruct an LLM for novel tasks are in-context learning and fine-tuning. To determine which is the most effective strategy, we apply these two strategies to the LLMs of the chain-of-thought reasoning, task planning, and motion planning modules respectively, benchmarking them on the downstream motion planning performance. As indicated in Table 5, in-context learning performs slightly better than fine-tuning in collision rates for reasoning and task planning, suggesting that in-context learning is a favorable choice in these modules. In motion planning, the fine-tuning strategy significantly outperforms in-context learning, demonstrating the necessity of fine-tuning LLMs in motion planning.

| Modules | | Avg. L2 (m) | Avg. Col. (%) |
|---|---|---|---|
| CoT Reason.+Task Plan. | Motion Plan. | | |
| Fine-tuning | In-context learning | 1.81 | 0.79 |
| In-context learning | In-context learning | 1.90 | 0.79 |
| Fine-tuning | Fine-tuning | 0.72 | 0.22 |
| In-context learning | Fine-tuning | 0.74 | 0.21 |

Table 5: **In-context learning vs. fine-tuning.** In-context learning performs slightly better in reasoning and task planning. Fine-tuning is indispensable for motion planning.

## 5 Conclusion

This work introduces Agent-Driver, a novel human-like paradigm that fundamentally transforms autonomous driving pipelines. Our key insight is to leverage LLMs as an agent to schedule different modules in autonomous driving. On top of the LLMs, we propose a tool library, a cognitive memory, and a reasoning engine to bring human-like intelligence into driving systems. Extensive experiments on the real-world driving dataset substantiate the effectiveness, few-shot learning ability, and interpretability of Agent-Driver. These findings shed light on the potential of LLMs as an agent in human-level intelligent driving systems. For future works, we plan to optimize the LLMs for real-time inference.

## Ethics Statement

This paper presents Agent-Driver, a novel method that leverages large language models as an agent for autonomous driving. While Agent-Driver uses language generation as part of its reasoning and planning process, the goal is to improve the quality of representations and decision-making for autonomous vehicles, not text generation itself. Therefore, the potential negative impact of Agent-Driver on areas like misinformation or deepfakes is minimal. On the privacy protection side, Agent-Driver uses driving data that does not include any personal or location information. In summary, we believe Agent-Driver does not lead to any ethics concerns.

## Reproduciblity Statement

We provide as many implementation details as possible in the paper submission and in the appendix. We also include the code for Agent-Driver, including the large language model, neural modules, tool library, cognitive memory, and reasoning engine components in the supplementary material. This will allow others to faithfully reproduce our results and build upon our proposed approach.

## Acknowledgments

We'd like to acknowledge Xinshuo Weng for fruitful discussions. We also acknowledge a gift from Google Research.

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

## Appendix

# A  Tool Library

In this section, we will first introduce the detailed descriptions of all functions in the tool library (Section A.1). Next, we will provide a detailed example of how the agent interacts with the tool library (Section A.2).

## A.1  Functions

We include all function definitions in the tool library in Tables 6 and 7. The proposed functions cover detection, prediction, occupancy, and mapping, and enable flexible and versatile environmental information collection.

## A.2  Tool Use

A detailed example of how the LLM leverages tools to collect environmental information is shown in Figures 4 and 5. System prompts are shown in blue, the response of LLMs is shown in green, and the collected data is shown in orange. System prompts provide sufficient context and guidance for instructing the LLM to dynamically invoke the functions in the tool library to collect necessary environmental information.

# B  Cognitive Memory

In this section, we detail the data format and retrieving process of the commonsense and experience memory.

## B.1  Memory Data

As shown in Figure 8 of the main text, the commonsense memory consists of essential knowledge for safe driving, which is cached in a text-based format and is fully configurable.

We build the experience memory by caching the environmental information of driving scenarios and corresponding driving trajectories in the training set. Please note that this experience memory can be editable online, meaning that expert demonstrations conducted by human drivers can be easily inserted into the memory and benefit the subsequent decision-making of Agent-Driver.

## B.2  Memory Search

We propose a two-stage searching strategy for retrieving the most similar past driving scenario to the query scenario.

In the first stage, we generate a vectorized key $k_i \in \mathbb{R}^{1 \times (n_e + n_g + n_h)}$ for each past scenario $i$ by vectorizing its ego-states $e_i \in \mathbb{R}^{1 \times n_e}$, mission goals $g_i \in \mathbb{R}^{1 \times n_g}$, and historical trajectories $h_i \in \mathbb{R}^{1 \times n_h}$. The $N$ past scenarios in the experience memory collectively construct a key tensor $K \in \mathbb{R}^{N \times (n_e + n_g + n_h)}$:

$$K = \{[e_i, g_i, h_i] | i = \{1, 2, ..., N\}\}. \tag{1}$$

Similarly, we can vectorize the query scenario into $Q = [e, g, h] \in \mathbb{R}^{1 \times (n_e + n_g + n_h)}$.

Subsequently, we compute the similarity scores $S \in \mathbb{R}^N$ between the querying scenario $Q$ and the past scenarios $K$:

$$S = Q \Lambda K^\top, \tag{2}$$

where $\Lambda = \text{diag}(\lambda_e, \lambda_g, \lambda_h) \in \mathbb{R}^{(n_e + n_g + n_h) \times (n_e + n_g + n_h)}$ indicates the weights of different components.

Finally, top-K samples with the K highest similarity scores are selected as candidates for the second-stage search.

In the second stage, we propose an LLM-based fuzzy search. The top-K past driving scenarios selected in the first stage are provided to the LLM. Then, the LLM is tasked to understand the text descriptions of these scenarios and determine the most similar past driving scenario to the query scenario. The driving trajectory corresponding to the selected scenario is also retrieved for reference.

With the proposed vector search and LLM-based fuzzy search, Agent-Driver can effectively retrieve the most similar past driving experience. The past experience and driving decision could help the current decision-making process.

## C  Reasoning Engine

In this section, we provide detailed information on the workflow of the reasoning engine. The reasoning engine takes environmental information and memory data as inputs, performs chain-of-thought reasoning, task planning, motion planning, and self-reflection, and eventually generates a driving trajectory for execution. Figures 9, 10, and 11 show an example of how the reasoning engine works. We denote the input environmental information as $\mathcal{O}$ and the retrieved memory data as $\mathcal{M}$. Please note that $\mathcal{O}$ and $\mathcal{M}$ are in the text format.

### C.1  Chain-of-Thought Reasoning

Chain-of-thought reasoning aims to emulate the human reasoning process and generate text-based reasoning results $\mathcal{R}$, which can be formulated as:

$$\mathcal{R} = F_{\text{LLM}}(\mathcal{O}, \mathcal{M}), \tag{3}$$

where $F_{\text{LLM}}$ is a LLM. To avoid arbitrary reasoning outputs of LLMs which might lead to hallucination and results not relevant to planning, we constrain $\mathcal{R}$ to contain two essential parts: notable objects and potential effects. Specifically, we first instruct the LLM to identify those notable objects that have critical impacts on decision-making from the input environmental information. Then, we instruct the LLM to assess how these notable objects will influence the subsequent decision-making process. The instruction can be established by two strategies: in-context learning and fine-tuning.

For in-context learning, each time we leverage four human-annotated examples $\mathcal{E}$ of notable objects and potential effects in addition to $\mathcal{O}$ and $\mathcal{M}$ collectively as inputs to the LLM:

$$\mathcal{R} = F_{\text{LLM}}(\mathcal{O}, \mathcal{M}, \mathcal{E}). \tag{4}$$

For fine-tuning, we auto-generate the reasoning targets $\hat{\mathcal{R}}$ leveraging the technique proposed in (Mao et al., 2023a). Then we fine-tune the LLM to make its reasoning outputs $\mathcal{R}$ approaching the targets $\hat{\mathcal{R}}$.

Both in-context learning and fine-tuning effectively reduce invalid outputs. As shown in Table 6 of the main paper, compared to the fine-tuning strategy, in-context learning enables the LLMs to generate more diverse reasoning outputs, and results in better motion planning performance.

### C.2  Task Planning

Task planning aims to generate high-level driving plans $\mathcal{P}$ for autonomous vehicles, taking the reasoning results $\mathcal{R}$ as well as the environmental information $\mathcal{O}$ and memory data $\mathcal{M}$ as inputs. The process can be formulated as

$$\mathcal{P} = F_{\text{LLM}}(\mathcal{O}, \mathcal{M}, \mathcal{R}). \tag{5}$$

We define the driving plan as a combination of discrete driving behaviors and speed estimations. In this paper, we proposed 6 discrete driving behaviors: `move_forward`, `change_lane_to_left`, `change_lane_to_right`, `turn_left`, `turn_right`, and `stop`. We also propose 6 speed estimations: `constant_speed`, `deceleration`, `quick_deceleration`,

`deceleration_to_zero`, `acceleration`, `quick_acceleration`. The combinations of driving behaviors and speed estimations result in 31 different driving plans (`stop` has no speed estimation). These driving plans can cover most driving scenarios and they are fully configurable, which means that we can add more behavior and speed types to cover those long-tailed scenarios. See Table 1 for details.

Similar to the reasoning module, in-context learning and fine-tuning can also be applied to instruct the LLM to generate driving plans. As shown in Table 6 of the main paper, in-context learning is more appropriate for instructing the LLM for task planning.

| Driving behavior | Speed estimation |
|---|---|
| `move_forward` | `constant_speed` |
| `change_lane_to_left` | `deceleration` |
| `change_lane_to_right` | `quick_deceleration` |
| `turn_left` | `deceleration_to_zero` |
| `turn_right` | `acceleration` |
| `stop` | `quick_acceleration` |

Table 1: Driving behaviors and speed estimations.

### C.3 Motion Planning

Motion planning aims to plan a safe and comfortable driving trajectory $\tau$, with the driving plan $\mathcal{P}$, reasoning results $\mathcal{R}$, environmental information $\mathcal{O}$, and memory data $\mathcal{M}$ as inputs. The process can be formulated as

$$\tau = F_{\text{LLM}}(\mathcal{O}, \mathcal{M}, \mathcal{R}, \mathcal{P}). \tag{6}$$

The planned trajectory $\tau$ can be represented as 6 waypoint coordinates in 3 seconds: $\tau = [(x_1, y_1), \cdots, (x_6, y_6)]$. A recent finding (Mao et al., 2023a) suggests a fine-tuned LLMs can generate text-based coordinates quite accurately. That is, the LLM can generate a text string "(1.23, 0.32)" representing a coordinate, and this can be easily transformed back into its numerical format $(1.23, 0.32)$ for subsequent execution. In particular, given a trajectory $\tau$, we first transform it into a sequence of language tokens $w$ using a tokenizer $T$:

$$\tau = T(\{(x_1, y_1), \cdots, (x_6, y_6)\}) = \{w_1, \cdots, w_n\}. \tag{7}$$

With these language tokens, we then reformulate motion planning as a language modeling problem:

$$\mathcal{L}_{\text{LM}} = -\sum_{i=1}^{N} \log P(\hat{w}_i | w_1, \cdots, w_{i-1}), \tag{8}$$

where $w$ and $\hat{w}$ are the language tokens of the planned trajectory $\tau$ from the LLM and the human driving trajectory $\hat{\tau}$ respectively. By learning to maximize the occurrence probability $P$ of the tokens $\hat{w}$ derived from the human driving trajectory $\hat{\tau}$, the LLM can generate human-like driving trajectories. We suggest readers refer to (Mao et al., 2023a) for more details.

With proper fine-tuning, the LLM is able to generate a text-based trajectory that can be further transformed into its numerical format. This step maps natural-language-based perception, memory, and reasoning into executable driving trajectories, enabling our agent to perform low-level actions.

### C.4 Self-Reflection

Self-reflection is designed to reassess and rectify the driving trajectory $\tau$ planned by the LLM. Specifically, the `collision_check` function is first invoked to check the collision of $\tau$ utilizing the estimated occupancy map. Specifically, we place the ego-vehicle at each waypoint in a trajectory, and then we will mark the trajectory as collision if there is an obstacle within a safe margin $\eta$ to the ego-vehicle in the occupancy map. If a trajectory is not marked with collision, we directly use this trajectory as output without further rectification. Otherwise, for those trajectories that have collisions, we leverage an optimization approach to rectify the trajectory $\tau$ into a collision-free one $\tau^*$. Following (Hu et al., 2023), we sample the obstacle points $O_t$ near each waypoint at timestep $t$ in the occupancy map. Then, an

optimization problem is formulated and solved through the Newton iteration method:

$$\tau^* = \arg\min_{\tau}(\|\tau - \tau^*\|_2 +$$

$$\sum_t \sum_{(x,y) \in O_t} \frac{\lambda}{\sigma\sqrt{2\pi}} \exp\left(-\frac{\|\tau_t^* - (x,y)\|_2^2}{2\sigma^2}\right)) \quad , \tag{9}$$

where $\lambda$ and $\sigma$ are hyperparameters. The first term regulates the optimized trajectory $\tau^*$ to be similar to the original $\tau$, and the second term pushes the waypoint $\tau_t^*$ in the trajectory away from the obstacle points $O_t$ for each timestep $t$.

Attributing to self-reflection, those unreliable decisions made by the LLM can further be corrected, and collisions in the planned trajectories can be effectively mitigated.

### C.5 Comparison with GPT-Driver

The most relevant implementation is GPT-Driver (Mao et al., 2023a), which handles the planning problem in autonomous driving by reformulating motion planning as a language modeling problem and introducing fine-tuned LLMs as a motion planner. Compared to (Mao et al., 2023a), our Agent-Driver introduce additional modules such as tool libraries and cognitive memories that GPT-Driver does not use. Also, it decouples the chain-of-thought reasoning, task planning, and motion planning as separate steps to the LLM. Compared to generating both reasoning and planning in a single pass as in GPT-Driver, we empirically found that such separation eases the learning and prediction process, and yields a better result. In addition, we use in-context learning to instruct the LLM to perform reasoning and task planning, which encourages better diversity and generalization ability. While in GPT-Driver, it uses fine-tuning. Finally, we additionally introduce a self-reflection stage to further rectify the motion planning results, which leads to fewer collisions and better safety.

## D  Experiments

### D.1  Evaluation Metrics

#### D.1.1  Open-Loop Metrics

As argued in (Hu et al., 2023), autonomous driving systems should be optimized in pursuit of the ultimate goal, *i.e.*, planning of the self-driving car. Hence, we focus on the motion planning performances to evaluate the effectiveness of our system. There are two commonly adopted metrics for motion planning on the nuScenes dataset: L2 error (in meters) and collision rate (in percentage). The average L2 error is computed by measuring each waypoint's distance in the planned and ground-truth trajectories, reflecting the proximity of a planned trajectory to a human driving trajectory. The collision rate is calculated by placing an ego-vehicle box on each waypoint of the planned trajectory and then checking for collisions with the ground truth bounding boxes of other objects, reflecting the safety of a planned trajectory. We follow the common practice and evaluate the motion planning result in a 3-second time horizon.

We further note that in different papers there are subtle discrepancies in computing these two metrics. For instance, in UniAD (Hu et al., 2023) both metrics at $k$-th second are measured as the error or collision rate at this certain timestep, while in ST-P3 (Hu et al., 2022) and following works (Jiang et al., 2023; Mao et al., 2023a), these metrics at $k$-th second is an average over $k$ seconds. There are also differences in ground truth objects for collision calculation in different papers. We detail the two different metric implementations as follows.

The output trajectory $\tau$ is formatted as 6 waypoints in a 3-second horizon, *i.e.*, $\tau = [(x_1, y_1), (x_2, y_2), (x_3, y_3), (x_4, y_4), (x_5, y_5), (x_6, y_6)]$. For $\tau$ in each driving scenario, the L2 error is computed as:

$$l_2 = \sqrt{(\tau - \hat{\tau})^2} = \left[\sqrt{(x_i - \hat{x}_i)^2 + (y_i - \hat{y}_i)^2}\right]_{i=1}^6 \quad , \tag{10}$$

where $l_2 \in \mathbb{R}^{6 \times 1}$ and $\hat{\tau}$ denotes human driving trajectory. Then, the average L2 error $\bar{l}_2 \in \mathbb{R}^{6 \times 1}$ can be computed by averaging $l_2$ for each sample in the test set.

In the UniAD metric (Hu et al., 2023), the L2 error at the $k$-th second ($k = 1, 2, 3$) is reported as the error at this timestep:

$$L_{2,k}^{\text{uniad}} = \bar{l}_2[2k]. \tag{11}$$

The average L2 error is then computed by averaging $L_{2,k}^{\text{uniad}}$ of the 3 timesteps.

In the ST-P3 metric (Hu et al., 2022) and following works (Jiang et al., 2023; Mao et al., 2023a), the L2 error at the $k$-th second is reported as the average error from 0 to $k$ second:

$$L_{2,k}^{\text{stp3}} = \frac{\sum_{t=1}^{2k} \bar{l}_2[t]}{2k}. \tag{12}$$

The average L2 error is computed by averaging the $L_{2,k}^{\text{stp3}}$ of the 3 timesteps again (average over average in other words).

In terms of the collision, it is computed by counting the number of times a planned trajectory collides with other objects in the ground truth occupancy map for all scenarios in the test set. We denote $\mathcal{C} \in \mathbb{N}^{6 \times 1}$ as the total collision times at each timestep.

Similarly, UniAD reports the collision $\mathcal{C}_k^{\text{uniad}}$ at the $k$-th second ($k = 1, 2, 3$) as $\mathcal{C}[2k]$, while ST-P3 reports $\mathcal{C}_k^{\text{stp3}}$ as the average from 0 to $k$ second:

$$\mathcal{C}_k^{\text{stp3}} = \frac{\sum_{t=1}^{2k} \mathcal{C}[t]}{2k}. \tag{13}$$

In addition to the differences in calculation methods, there is also a difference in how the ground truth occupancy maps are generated in the two metrics. Specifically, UniAD only considers the vehicle category when generating ground truth occupancy, while ST-P3 considers both the vehicle and pedestrian categories. This difference leads to varying collision rates for identical planned trajectories when measured by these two metrics, yet it does not impact the L2 error.

In this paper, we faithfully evaluated our approach and the baseline methods using the officially implemented evaluation metrics in the two papers (Hu et al., 2023; 2022), ensuring a completely fair comparison with other methods.

### D.1.2 Closed-Loop Metrics

Within the CARLA simulator (Dosovitskiy et al., 2017), the metrics of route completion and driving score are generally used to evaluate the planning performance of autonomous driving systems.

Route completion is a straightforward metric, measuring the percentage of a predefined route that the autonomous vehicle successfully completes before the simulation ends. This metric is crucial for understanding the basic capability of an autonomous vehicle to navigate from point A to point B.

The driving score is calculated by adjusting the route completion rate with a penalty factor that considers various infractions, including collisions with pedestrians, other vehicles, and stationary objects, deviations from the planned route, lane violations, ignoring red traffic lights, and failing to stop at stop signs.

### D.2 Implementation Details

We employ gpt-3.5-turbo-0613 as the foundation LLM in our Agent-Driver. We leverage the same LLM for every task except motion planning, and we fine-tuned another LLM specially for motion planning.

| ID | Tool Library | Common. Memory | Exp. Memory | CoT Reason. | Task Plan. | Self-Reflect. | L2 (m) ↓ | | | | Collision (%) ↓ | | | |
|----|------|------|------|------|------|------|------|------|------|------|------|------|------|------|
| | | | | | | | 1s | 2s | 3s | Avg. | 1s | 2s | 3s | Avg. |
| 1 | ✗ | ✓ | ✓ | ✓ | ✓ | ✗ | 0.24 | 0.71 | 1.44 | 0.80 | 0.03 | 0.27 | 0.91 | 0.40 |
| 2 | ✓ | ✗ | ✓ | ✓ | ✓ | ✗ | 0.24 | 0.69 | 1.42 | 0.79 | 0.03 | 0.23 | 0.83 | 0.37 |
| 3 | ✓ | ✓ | ✗ | ✓ | ✓ | ✗ | 0.24 | 0.72 | 1.46 | 0.81 | 0.07 | 0.23 | 0.96 | 0.42 |
| 4 | ✓ | ✓ | ✓ | ✗ | ✓ | ✗ | 0.24 | 0.71 | 1.45 | 0.80 | 0.03 | 0.23 | 0.88 | 0.38 |
| 5 | ✓ | ✓ | ✓ | ✓ | ✗ | ✗ | 0.25 | 0.72 | 1.47 | 0.81 | 0.05 | 0.23 | 0.93 | 0.40 |
| 6 | ✓ | ✓ | ✓ | ✓ | ✓ | ✗ | 0.24 | 0.70 | 1.42 | 0.79 | 0.03 | 0.20 | 0.81 | 0.35 |
| 7 | ✓ | ✓ | ✓ | ✓ | ✓ | ✓ | 0.25 | 0.71 | 1.43 | 0.80 | 0.03 | 0.08 | 0.56 | 0.23 |

Table 2: **Ablation of components in Agent-Driver.** The removal of any component can influence the planning efficacy of our system, indicating the importance of all components in our system.

For tool use and memory search, the LLM is guided by system prompts without fine-tuning or exemplar-based in-context learning. In chain-of-thought reasoning and task planning, the LLM is instructed by two randomly selected exemplars derived from the training set. This approach encourages the models to develop various insightful chain-of-thought processes and detailed plans for tasks independently. Please note that we utilize the original pre-trained LLM with different system prompts and user input for the above tasks. In motion planning, we fine-tune an LLM with human driving trajectories for only *one epoch*.

## D.3   Ablation Study

Table 2 shows the results of ablating different components in Agent-Driver. All variants utilize 10% training data for instructing the LLMs. From ID 1 to ID 5, we ablate the main components in Agent-Driver, respectively. We deactivate the self-reflection module and directly evaluate the trajectories output from LLMs to better assess the contribution of each other module. When the tool library is disabled, all perception results form the input to Agent-Driver without selection, which yields ∼2 times more input tokens and harms the system's efficiency. The removal of the tool library also increases the collision rate, indicating the effectiveness of this component. In addition, the collision rate gets worse by removing the commonsense and experience memory, reasoning, and task planning modules, demonstrating the necessity of these components. Besides, we further note that self-reflection also greatly reduces the collision rate.

## D.4   Impact of Ego-States on Open-Loop Motion Planning

As pointed out by Zhai et al. (2023) and Li et al. (2023), ego status information can provide a strong heuristic for open-loop motion planning. To investigate the impact of ego states on Agent-Driver, we conduct experiments of Agent-Driver using purely ego-status and without ego-status. The results are shown in Table 3, where all settings are trained with 10% data. We can see that: (1) The ego-only baseline is worse than

| Settings | Avg. L2 (m) | Avg. Col. (%) |
|----------|-------------|---------------|
| Ego-only | 0.93 | 0.32 |
| Agent-Driver w/o ego | 1.00 | 0.32 |
| UniAD | 1.03 | 0.31 |
| Agent-Driver | **0.80** | **0.23** |

Table 3: Impact of ego-states on Agent-Driver.

Agent-Driver, which indicates Agent-Driver does not merely rely on ego-states but also benefits from the proposed components, and (2) After dropping the ego-states, Agent-Driver still works well and performs on-par with UniAD, which means Agent-Driver can still work decently well without ego-states.

We further note that most prior works that Agent-Driver compared with in Table 2 of the main text use ego-states for planning. Specifically, VAD explicitly inputs ego-states as side information for the planner, and UniAD implicitly includes ego-states via its BEV module (Li et al., 2023).

## D.5   Compatibility with Different Neural Modules

As shown in Table 4, Agent-Driver constantly maintains a favorable performance with combinations of variable neural modules. We argue that discrepancy in perception and prediction performance can be compensated by strong reasoning systems and is no longer the bottleneck of our system. Notably, unlike conventional frame-

| Neural Modules | | | | Avg. L2 (m) | Avg. Col. (%) |
|---|---|---|---|---|---|
| Detection | Prediction | Occupancy | Mapping | | |
| VAD | VAD | ST-P3 | ST-P3 | 0.73 | 0.24 |
| VAD | VAD | UniAD | UniAD | 0.72 | 0.22 |
| UniAD | UniAD | ST-P3 | ST-P3 | 0.74 | 0.24 |
| UniAD | UniAD | UniAD | UniAD | 0.74 | 0.21 |

Table 4: Compatibility to different perception modules.

works which need retraining upon any module change, attributed to the flexibility of Agent-Driver, all neural modules in our system can be displaced in a *plug-and-play* manner, indicating our system's compatibility.

## D.6  Language Justification on the BDD-X Dataset

Our method is compatible with diverse functional calls in the tool library. By incorporating the state-of-the-art vision-language model (Jin et al., 2023) as a tool, our Agent-Driver can provide language justification to actions based on visual inputs. We demonstrate this capability on the BDD-X dataset, which is a commonly used language benchmark for autonomous driving. Both quantitative results in Table 5 and qualitative results in Figure 1 have demonstrated the language justification ability of Agent-Driver.

| Method | Narration | | | | Reasoning | | | |
|---|---|---|---|---|---|---|---|---|
| | BLEU4 | METEOR | ROUGE-L | CIDEr | BLEU4 | METEOR | ROUGE-L | CIDEr |
| Agent-Driver | 34.55 | 30.58 | 62.75 | 246.79 | 11.65 | 15.36 | 32.00 | 103.16 |

Table 5: Quantitative performance of Agent-Driver on the BDD-X dataset.

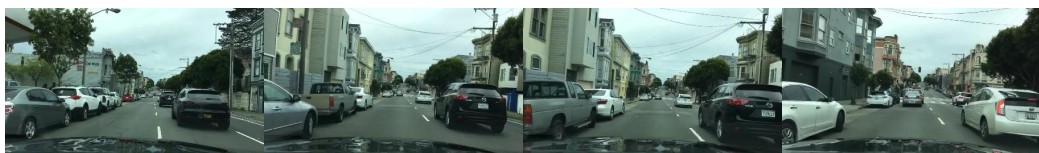

**Action description:** the car is driving forward.
**Action justification:** because the path is clear.

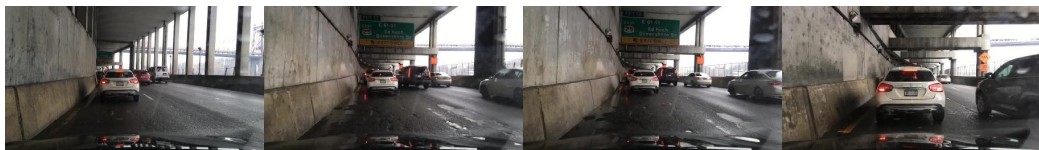

**Action description:** the car is driving slowly down the road.
**Action justification:** because traffic ahead is moving slowly.

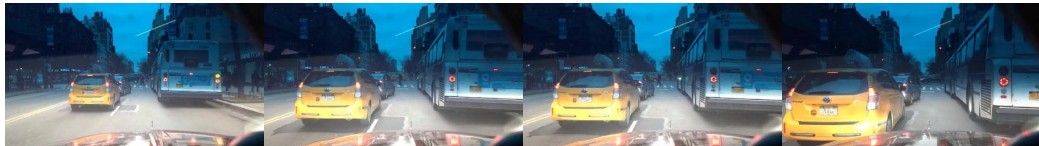

**Action description:** the car slows down.
**Action justification:** because the car in front of it has stopped.

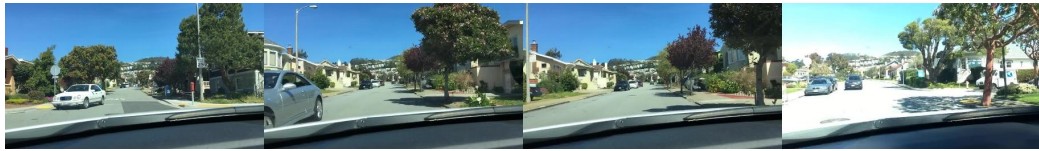

**Action description:** the car is driving forward.
**Action justification:** because there are no other cars on the road.

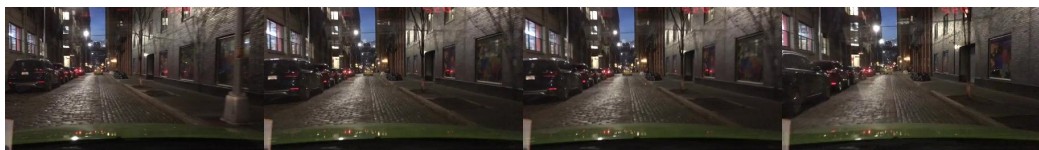

**Action description:** the car is driving forward.
**Action justification:** because there are no other cars on the road.

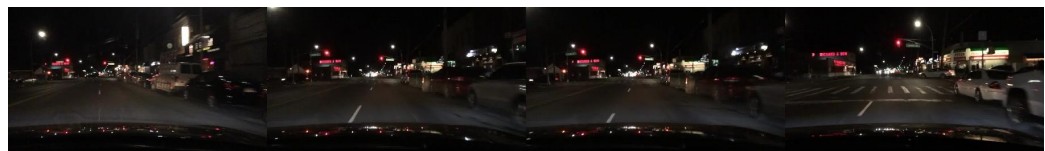

**Action description:** the car is driving forward.
**Action justification:** because there is no traffic in that lane.

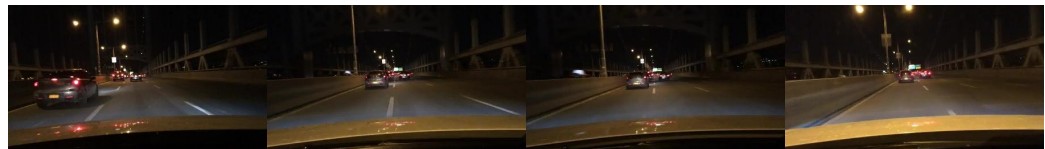

**Action description:** the car merges left.
**Action justification:** because the lane is clear.

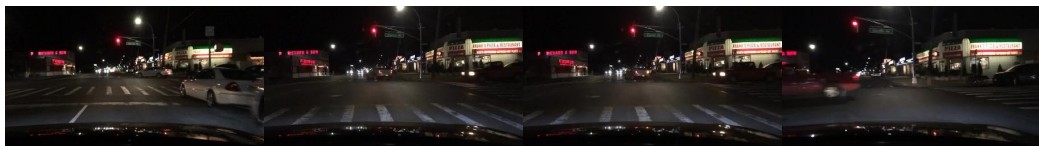

**Action description:** the car is slowing down to a complete stop.
**Action justification:** because the traffic light at the intersection has turned red.

Figure 1: Visualization of Agent-Driver language justification capability on BDD-X.

# E  Qualitative Analysis

## E.1  Qualitative Ablation

A qualitative visualization of ablating the system components in Agent-Driver is shown in Figure 2. The result shows that when ablating the memory module of Agent-Driver, the planned trajectory (in purple) has a larger discrepancy to the ground truth trajectory. When deactivating the chain-of-thought reasoning and task-level planning, the planned trajectory (in yellow) deviates more from the ground truth, suggesting that reasoning is a critical component for accurate planning of the system. The qualitative ablation further verifies the effectiveness of the proposed components in our Agent-Driver.

### E.2 Qualitative Results

Figure 12 provides more examples to show the interpretability of Agent-Driver. Figures 13, 14 and 15 visualize critical objects identified by Agent-Driver and the planned driving trajectories, from which we can see our system progressively identifies critical objects via tool use and reasoning, and eventually plans a safe trajectory for driving. Driving videos are shown in Figures 16, 17, and 18. The qualitative results verify the effectiveness and interpretability of our Agent-Driver.

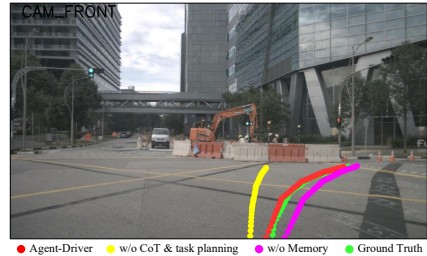

Figure 2: Qualitative ablation of system components.

### E.3 Failure Cases

We provide failure cases in Agent-Driver in Figure 3. We observed that heading errors of large objects, *e.g.*, buses, have a critical impact on motion planning, which indicates the importance of accurate heading prediction in detection networks.

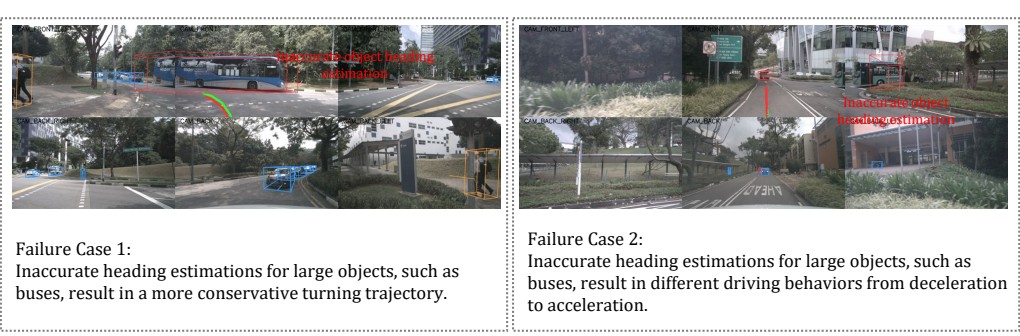

Failure Case 1:
Inaccurate heading estimations for large objects, such as buses, result in a more conservative turning trajectory.

Failure Case 2:
Inaccurate heading estimations for large objects, such as buses, result in different driving behaviors from deceleration to acceleration.

Figure 3: Failure cases in Agent-Driver.

## F   Limitations

Due to the limitations of the OpenAI APIs, we are unable to obtain the accurate inference time of our Agent-Driver. Thus it remains uncertain whether our approach can meet the real-time demands of commercial driving applications. However, we argue that recent advances in accelerating LLM inference (Dao et al., 2022; Liu et al., 2023; Xiao et al., 2023) shed light on a promising direction to enable LLMs for real-time applications. In the meantime, due to hardware development, autonomous vehicles' onboard computational power also evolves rapidly. Given these two factors, we believe the inference restriction will be resolved in the near future.

| Task | Function Name | Function Descriptions | Parameters |
|---|---|---|---|
| Detection | get_leading_object_detection | Get the detection of the leading object, the function will return the leading object id and its position and size. If there is no leading object, return None. | Null |
| | get_surrounding_object_detections | Get the detections of the surrounding objects in a 20m*20m range, the function will return a list of surrounding object ids and their positions and sizes. If there is no surrounding object, return None. | Null |
| | get_front_object_detections | Get the detections of the objects in front of you in a 20m*40m range, the function will return a list of front object ids and their positions and sizes. If there is no front object, return None. | Null |
| | get_object_detections_in_range | Get the detections of the objects in a customized range (x_start, x_end)*(y_start, y_end)m$^2$, the function will return a list of object ids and their positions and sizes. If there is no object, return None. | ["x_start", "x_end", "y_start", "y_end"] |
| | get_all_object_detections | Get the detections of all objects in the whole scene, the function will return a list of object ids and their positions and sizes. Always avoid using this function if there are other choices. | Null |
| Prediction | get_leading_object_future_trajectory | Get the predicted future trajectory of the leading object, the function will return a trajectory containing a series of waypoints. If there is no leading vehicle, return None. | Null |
| | get_future_trajectories_for_specific_objects | Get the future trajectories of specific objects (specified by a List of object ids), the function will return trajectories for each object. If there is no object, return None. | ["object_ids"] |
| | get_future_trajectories_in_range | Get the future trajectories where any waypoint in this trajectory falls into a given range (x_start, x_end)*(y_start, y_end)m$^2$, the function will return each trajectory that satisfies the condition. If there is no trajectory satisfied, return None | ["x_start", "x_end", "y_start", "y_end"] |
| | get_future_waypoint_of_specific_objects_at_timestep | Get the future waypoints of specific objects at a specific timestep, the function will return a list of waypoints. If there is no object or the object does not have a waypoint at the given timestep, return None. | ["object_ids", "timestep"] |
| | get_all_future_trajectories | Get the predicted future trajectories of all objects in the whole scene, the function will return a list of object ids and their future trajectories. Always avoid using this function if there are other choices. | Null |

Table 6: Function definitions in the tool library.

| Task | Function Name | Function Descriptions | Parameters |
|---|---|---|---|
| Map | get_drivable_at_locations | Get the drivability at the locations $[(x_1, y_1), ..., (x_n, y_n)]$. If the location is out of the map scope, return None. | ["locations"] |
| | check_drivable_of_planned_trajectory | Check the drivability at the planned trajectory. | ["trajectory"] |
| | get_lane_category_at_locations | Get the lane category at the locations $[(x_1, y_1), ..., (x_n, y_n)]$. If the location is out of the map scope, return None. | ["locations", "ret_prob"] |
| | get_distance_to_shoulder_at_locations | Get the distance to both sides of road shoulders at the locations $[(x_1, y_1), ..., (x_n, y_n)]$. If the location is out of the map scope, return None. | ["locations"] |
| | get_current_shoulder | Get the distance to both sides of road shoulders for the current ego-vehicle location. | Null |
| | get_distance_to_lane_divider_at_locations | Get the distance to both sides of road lane dividers at the locations $[(x_1, y_1), ..., (x_n, y_n)]$. If the location is out of the map scope, return None. | ["locations"] |
| | get_current_lane_divider | Get the distance to both sides of road lane dividers for the current ego-vehicle location. | Null |
| | get_nearest_pedestrian_crossing | Get the location of the nearest pedestrian crossing to the ego-vehicle. If there is no such pedestrian crossing, return None. | Null |
| Occupancy | get_occupancy_at_locations_for_timestep | Get the probability whether a list of locations $[(x_1, y_1), ..., (x_n, y_n)]$ is occupied at the timestep t. If the location is out of the occupancy prediction scope, return None. | ["locations", "timestep"] |
| | check_collision_for_planned_trajectory | Check the probability of whether a planned trajectory $[(x_1, y_1), ..., (x_n, y_n)]$ collides with other objects. | ["trajectory"] |

Table 7: Function definitions in the tool library. Cont'd.

**A Language Agent for Autonomous Driving**
Role: You are the brain of an autonomous vehicle (a.k.a. ego-vehicle). In this step, you need to extract necessary information from the driving scenario. The information you extracted must be useful to the next-step motion planning.

Necessary information might include the following:
- Detections: The detected objects that you need to pay attention to.
- Predictions: The estimated future motions of the detected objects.
- Maps: Map information includes traffic lanes and road boundaries.
- Occunpancy: Occupancy implies whether a location has been occupied by other objects.

Task
- You should think about what types of information (Detections, Predictions, Maps, Occupancy) you need to extract from the driving scenario.
- Detections and Predictions are quite important for motion planning. You should call at least one of them if necessary.
- Maps information are also important. You should pay more attention to road shoulder and lane divider information to your current ego-vehicle location.
- I will guide you through the thinking process step by step.

*****Context Information:*****
Current State:
 - Velocity (vx,vy): (-0.01,0.92)
 - Heading Angular Velocity (v_yaw): (0.00)
 - Acceleration (ax,ay): (-0.00,-0.50)
 - Can Bus: (-0.74,0.14)
 - Heading Speed: (0.95)
 - Steering: (-0.02)
Historical Trajectory (last 2 seconds): [(-0.07,-6.43), (-0.05,-4.34), (-0.02,-2.32), (-0.01,-0.91)]
Mission Goal: FORWARD

Do you need to perform detections in this driving scenario?
Please answer YES or NO.

YES

You can execute one of the following functions to get object detection results (don't execute functions that have been used before):
- get_leading_object_detection() #Get the detection of the leading object, the function will return the leading object id and its position and size. If there is no leading object, return None
- get_object_detections_in_range(x_start, x_end, y_start, y_end) #Get the detections of the objects in a given range (x_start, x_end)*(y_start, y_end)m^2, the function will return a list of object ids and their positions and sizes. If there is no object, return None
- get_surrounding_object_detections() #Get the detections of the surrounding objects in a 20m*20m range, the function will return a list of surroundind object ids and their positions and sizes. If there is no surrounding object, return None
- get_front_object_detections() #Get the detections of the objects in front of you in a 10m*20m range, the function will return a list of front object ids and their positions and sizes. If there is no front object, return None
- get_all_object_detections() #Get the detections of all objects in the whole scene, the function will return a list of object ids and their positions and sizes. Always avoid using this function if there are other choices.

{"name": "get_front_object_detections", "arguments": "{}"}

Front object detections:
Front object detected, object type: car, object id: 2, position: (4.36, 9.56), size: (1.86, 4.72)
Front object detected, object type: car, object id: 3, position: (-3.70, 13.08), size: (2.01, 4.92)

Do you need to perform future trajectory predictions for the detected objects?
Please answer YES or NO.

YES

Figure 4: An example of the tool use process.

You can execute one of the following functions to get object future trajectory predictions (don't execute functions that have been used before):
- get_leading_object_future_trajectory() #Get the predicted future trajectory of the leading object, the function will return a trajectory containing a series of waypoints. If there is no leading vehicle, return None
- get_future_trajectories_for_specific_objects(object_ids) #Get the future trajectories of specific objects (specified by a List of object ids), the function will return trajectories for each object. If there is no object, return None
- get_future_trajectories_in_range(x_start, x_end, y_start, y_end) #Get the future trajectories where any waypoint in this trajectory falls into a given range (x_start, x_end)*(y_start, y_end)m^2, the function will return each trajectory that satisfies the condition. If there is no trajectory satisfied, return None
- get_future_waypoint_of_specific_objects_at_timestep(object_ids, timestep) #Get the future waypoints of specific objects at a specific timestep, the function will return a list of waypoints. If there is no object or the object does not have a waypoint at the given timestep, return None
- get_all_future_trajectories() #Get the predicted future trajectories of all objects in the whole scene, the function will return a list of object ids and their future trajectories. Always avoid using this function if there are other choices.

{"name": "get_future_trajectories_for_specific_objects", "arguments": "'object_ids': [2, 3]"}

Future trajectories for specific objects:
Object type: car, object id: 2, future waypoint coordinates in 3s: [(4.36, 9.56), (4.36, 9.56), (4.36, 9.57), (4.36, 9.57), (4.36, 9.56), (4.36, 9.56)]
Object type: car, object id: 3, future waypoint coordinates in 3s: [(-2.66, 13.82), (-1.69, 14.79), (-0.99, 16.13), (-0.25, 17.73), (0.19, 19.42), (0.57, 21.35)]

Do you need to get occupancy information for this driving scenario?
Please answer YES or NO.

NO

Do you need to get map information for this driving scenario?
Please answer YES or NO.

YES

You can execute one of the following functions to get map information (don't execute functions that have been used before):
- get_drivable_at_locations(locations) #Get the drivability at the locations [(x_1, y_1), ..., (x_n, y_n)]. If the location is out of the map scope, return None
- get_lane_category_at_locations(locations, return_score) #Get the lane category at the locations [(x_1, y_1), ..., (x_n, y_n)]. If the location is out of the map scope, return None
- get_distance_to_shoulder_at_locations(locations) #Get the distance to both sides of road shoulders at the locations [(x_1, y_1), ..., (x_n, y_n)]. If the location is out of the map scope, return None
- get_current_shoulder() #Get the distance to both sides of road shoulders for the current ego-vehicle location.
- get_distance_to_lane_divider_at_locations(locations) #Get the distance to both sides of road lane_dividers at the locations [(x_1, y_1), ..., (x_n, y_n)]. If the location is out of the map scope, return None
- get_current_lane_divider() #Get the distance to both sides of road lane_dividers for the current ego-vehicle location
- get_nearest_pedestrian_crossing() #Get the location of the nearest pedestrian crossing to the ego-vehicle. If there is no such pedestrian crossing, return None

{"name": "get_current_shoulder", "arguments": "{}"}

Distance to both sides of road shoulders of current ego-vehicle location:
Current ego-vehicle's distance to left shoulder is 7.5m and right shoulder is 4.0m

Figure 5: An example of the tool use process. Cont'd.

**A Language Agent for Autonomous Driving**
Role: You are the brain of an autonomous vehicle (a.k.a. ego-vehicle). In this step, you need to retrieve the most similar past driving experience to help decision-making.

Task
- You will receive the current driving scenario.
- You will also receive several past driving experiences.
- You should decide ONLY ONE experience that is most similar to the current scenario based on the information provided.
- Please answer ONLY the index (e.g., 0, 1, 2) of the most similar experience.

**A Language Agent for Autonomous Driving**
Role: You are the brain of an autonomous vehicle (a.k.a. ego-vehicle). In this step, you need to retrieve the most similar past driving experience to help decision-making.

Task
- You will receive the current driving scenario.
- You will also receive several past driving experiences.
- You should decide ONLY ONE experience that is most similar to the current scenario based on the information provided.
- Please answer ONLY the index (e.g., 0, 1, 2) of the most similar experience.

## Current Scenario:
*****Environmental Information:*****
Current State:
 - Velocity (vx,vy): (-0.01,0.92)
 - Heading Angular Velocity (v_yaw): (0.00)
 - Acceleration (ax,ay): (-0.00,-0.50)
 - Can Bus: (-0.74,0.14)
 - Heading Speed: (0.95)
 - Steering: (-0.02)
Historical Trajectory (last 2 seconds): [(-0.07,-6.43), (-0.05,-4.34), (-0.02,-2.32), (-0.01,-0.91)]
Mission Goal: FORWARD

Front object detections:
Front object detected, object type: car, object id: 2, position: (4.36, 9.56), size: (1.86, 4.72)
Front object detected, object type: car, object id: 3, position: (-3.70, 13.08), size: (2.01, 4.92)

Future trajectories for specific objects:
Object type: car, object id: 2, future waypoint coordinates in 3s: [(4.36, 9.56), (4.36, 9.56), (4.36, 9.57), (4.36, 9.57), (4.36, 9.56), (4.36, 9.56)]
Object type: car, object id: 3, future waypoint coordinates in 3s: [(-2.66, 13.82), (-1.69, 14.79), (-0.99, 16.13), (-0.25, 17.73), (0.19, 19.42), (0.57, 21.35)]

Distance to both sides of road shoulders of current ego-vehicle location:
Current ego-vehicle's distance to left shoulder is 7.5m and right shoulder is 4.0m

Found 3 relevant experiences:

## Past Driving Experience 1:
*****Past Environmental Information:*****
Current State:
 - Velocity (vx,vy): (0.00,1.07)
 - Heading Angular Velocity (v_yaw): (-0.00)
 - Acceleration (ax,ay): (-0.02,-0.43)
 - Can Bus: (-0.67,0.03)
 - Heading Speed: (1.00)
 - Steering: (0.13)
Historical Trajectory (last 2 seconds): [(-0.16,-6.66), (-0.08,-4.46), (-0.03,-2.55), (-0.00,-1.06)]
Mission Goal: FORWARD

Future trajectories for specific objects:
Object type: car, object id: 2, future waypoint coordinates in 3s: [(-1.13, -13.82), (-1.09, -12.18), (-1.05, -10.66), (-0.98, -9.22), (-0.98, -7.96), (-0.93, -6.74)]
Object type: car, object id: 3, future waypoint coordinates in 3s: [(-25.19, -17.79), (-25.19, -17.79), (-25.18, -17.78), (-25.18, -17.78), (-25.18, -17.78), (-25.17, -17.78)]

Figure 6: An example of the memory search process.

## Past Driving Experience 2:
*****Past Environmental Information:*****
Current State:
 - Velocity (vx,vy): (-0.01,0.97)
 - Heading Angular Velocity (v_yaw): (0.00)
 - Acceleration (ax,ay): (-0.01,-0.46)
 - Can Bus: (-0.68,0.11)
 - Heading Speed: (1.16)
 - Steering: (0.04)
Historical Trajectory (last 2 seconds): [(-0.06,-6.03), (-0.04,-4.10), (-0.02,-2.40), (-0.01,-0.97)]
Mission Goal: FORWARD

Front object detections:
Front object detected, object type: car, object id: 0, position: (-0.58, 9.54), size: (1.75, 3.99)
Front object detected, object type: car, object id: 2, position: (3.57, 7.27), size: (1.88, 4.49)
Front object detected, object type: truck, object id: 3, position: (-0.08, 19.97), size: (2.93, 9.04)
Future trajectories for specific objects:
Object type: car, object id: 2, future waypoint coordinates in 3s: [(3.59, 12.47), (3.61, 17.78), (3.68, 23.26), (3.70, 28.79), (3.66, 34.46), (3.65, 40.07)]
Object type: truck, object id: 3, future waypoint coordinates in 3s: [(-0.08, 19.96), (-0.08, 19.93), (-0.09, 19.93), (-0.09, 19.92), (-0.08, 19.92), (-0.09, 19.91)]

## Past Driving Experience 3:
*****Past Environmental Information:*****
Current State:
 - Velocity (vx,vy): (-0.01,1.06)
 - Heading Angular Velocity (v_yaw): (-0.00)
 - Acceleration (ax,ay): (0.00,-0.32)
 - Can Bus: (-0.77,0.04)
 - Heading Speed: (1.06)
 - Steering: (-0.01)
Historical Trajectory (last 2 seconds): [(-0.08,-6.01), (-0.05,-4.12), (-0.04,-2.43), (-0.02,-1.06)]
Mission Goal: FORWARD

Front object detections:
Front object detected, object type: car, object id: 1, position: (-3.08, 2.56), size: (1.88, 4.52)
Future trajectories for specific objects:
Object type: car, object id: 2, future waypoint coordinates in 3s: [(-18.54, -24.30), (-18.54, -24.30), (-18.54, -24.30), (-18.54, -24.30), (-18.54, -24.31), (-18.54, -24.31)]
Object type: car, object id: 3, future waypoint coordinates in 3s: [(-3.26, -31.78), (-3.43, -28.27), (-3.36, -25.04), (-3.45, -22.14), (-3.46, -19.62), (-3.51, -17.45)]

Please return the index 1-3 of the most similar experience:

2

*****Past Driving Experience for Reference:*****
Most similar driving experience from memory with confidence score: 0.75:
The planned trajectory in this experience for your reference:
[(0.01,0.98), (0.01,1.87), (0.02,2.40), (0.02,2.64), (0.02,2.76), (0.02,2.86)]

Figure 7: An example of the memory search process. Cont'd.

*****Common Sense:*****
 - Avoid collision with other objects.
 - Always drive on drivable regions.
 - Avoid driving on occupied regions.
 - Pay attention to your ego-states and historical trajectory when planning.
 - Maintain a safe distance from the objects in front of you.

Figure 8: Retrieved commonsense memory.

**A Language Agent for Autonomous Driving**
Role: You are the brain of an autonomous vehicle (a.k.a. ego-vehicle). In this step, you need to first determine notable objects and identify their potential effects on your driving route, and then derive a high-level driving plan.

Context:
- Coordinates: X-axis is perpendicular, and Y-axis is parallel to the direction you're facing. You're at point (0,0). Units: meters.

Input
- You will receive your current ego-states.
- You will also receive current perception results.

Task
- You need to determine the notable objects based on perception results and ego-states. Notable objects are the objects that will have potential effects on your driving route. So you should always pay attention to the objects in front (with positive y) of you, and the objects that are close (within 1.5 meters) to you.
- You need to describe the potential effects of those notable objects on your driving route.
- You need to derive a high-level driving plan based on the former information and reasoning results.
The driving plan should be a combination of a meta action from ["STOP", "MOVE FORWARD", "TURN LEFT", "CHANGE LANE TO LEFT", "TURN RIGHT", "CHANE LANE TO RIGHT"], and a speed description from ["A CONSTANT SPEED", "A DECELERATION", "A QUICK DECELERATION", "A DECELERATION TO ZERO", "AN ACCELERATION", "A QUICK ACCELERATION"] if the meta action is not "STOP".
- **Strictly follow the output format.**

Output:
*****Chain-of-Thoughts Reasoning:*****
 - Notable Objects:
   Potential Effects:
 - Notable Objects:
   Potential Effects:
*****Task Planning:*****
Behavior: , Speed:
Driving plan:

---

Here are examples for your reference:

## Example 1
## Input:
*****Environmental Information:*****
Current State:
 - Velocity (vx,vy): (-0.02,2.66)
 - Heading Angular Velocity (v_yaw): (-0.01)
 - Acceleration (ax,ay): (0.00,0.00)
 - Can Bus: (-1.72,-0.95)
 - Heading Speed: (2.83)
 - Steering: (1.12)
Historical Trajectory (last 2 seconds): [(-1.16,-10.63), (-0.87,-7.97), (-0.58,-5.32), (-0.29,-2.66)]
Mission Goal: RIGHT

Front object detections:
Front object detected, object type: bicycle, object id: 0, position: (-1.02, 7.49), size: (0.49, 1.67)
Front object detected, object type: car, object id: 1, position: (8.71, 18.66), size: (1.92, 4.55)

Future trajectories for specific objects:
Object type: bicycle, object id: 0, future waypoint coordinates in 3s: [(-1.02, 7.51), (-1.02, 7.52), (-1.02, 7.54), (-1.03, 7.55), (-1.02, 7.59), (-1.02, 7.61)]
Object type: car, object id: 1, future waypoint coordinates in 3s: [(8.71, 18.66), (8.70, 18.65), (8.69, 18.65), (8.69, 18.64), (8.69, 18.63), (8.69, 18.65)]

Distance to both sides of road shoulders of current ego-vehicle location:
Current ego-vehicle's distance to left shoulder is 1.0m and right shoulder is 0.5m

## Expected Output:
*****Chain-of-Thoughts Reasoning:*****
 - Notable Objects: bicycle at (-1.02,7.49), moving to (-1.02,7.51) at 0.5 second
   Potential Effects: within the safe zone of the ego-vehicle at 0.5 second
*****Task Planning:*****
Behavior: TURN RIGHT, Speed: A CONSTANT SPEED
Driving plan: TURN RIGHT WITH A CONSTANT SPEED

Figure 9: An example of chain-of-thought reasoning and task planning.

## Example 2
## Input:
*****Environmental Information:*****
Current State:
 - Velocity (vx,vy): (-0.10,5.42)
 - Heading Angular Velocity (v_yaw): (-0.00)
 - Acceleration (ax,ay): (0.02,1.14)
 - Can Bus: (0.92,0.25)
 - Heading Speed: (4.53)
 - Steering: (0.03)
Historical Trajectory (last 2 seconds): [(-0.17,-17.86), (-0.11,-13.82), (-0.07,-9.70), (-0.04,-5.42)]
Mission Goal: FORWARD

Front object detections:
Front object detected, object type: pedestrian, object id: 4, position: (6.49, 16.88), size: (0.66, 0.72)

Future trajectories for specific objects:
Object type: pedestrian, object id: 4, future waypoint coordinates in 3s: [(6.46, 17.53), (6.44, 18.20), (6.42, 18.89), (6.38, 19.57), (6.37, 20.26), (6.34, 20.91)]

Distance to both sides of road shoulders of selected locations:
Location (6.49, 16.88) distance to left shoulder is 2.5m and distance to right shoulder is uncertain

## Expected Output:
*****Chain-of-Thoughts Reasoning:*****
 - Notable Objects: car at (2.44,44.97)
   Potential Effects: within the safe zone of the ego-vehicle at 2.5 second
*****Task Planning:*****
Behavior: MOVE FORWARD, Speed: A DECELERATION
Driving plan: MOVE FORWARD WITH A DECELERATION

*****Environmental Information:*****
Current State:
 - Velocity (vx,vy): (-0.01,0.92)
 - Heading Angular Velocity (v_yaw): (0.00)
 - Acceleration (ax,ay): (-0.00,-0.50)
 - Can Bus: (-0.74,0.14)
 - Heading Speed: (0.95)
 - Steering: (-0.02)
Historical Trajectory (last 2 seconds): [(-0.07,-6.43), (-0.05,-4.34), (-0.02,-2.32), (-0.01,-0.91)]
Mission Goal: FORWARD

Front object detections:
Front object detected, object type: car, object id: 2, position: (4.36, 9.56), size: (1.86, 4.72)
Front object detected, object type: car, object id: 3, position: (-3.70, 13.08), size: (2.01, 4.92)

Future trajectories for specific objects:
Object type: car, object id: 2, future waypoint coordinates in 3s: [(4.36, 9.56), (4.36, 9.56), (4.36, 9.57), (4.36, 9.57), (4.36, 9.56), (4.36, 9.56)]
Object type: car, object id: 3, future waypoint coordinates in 3s: [(-2.66, 13.82), (-1.69, 14.79), (-0.99, 16.13), (-0.25, 17.73), (0.19, 19.42), (0.57, 21.35)]

Distance to both sides of road shoulders of selected locations:
Location (4.36, 9.56) distance to left shoulder is 16.5m and right shoulder is 0.5m
Location (-3.70, 13.08) distance to left shoulder is 13.0m and right shoulder is 8.5m

*****Chain of Thoughts Reasoning:*****
- Notable Objects: car at (4.36,9.56)
  Potential Effects: within the safe zone of the ego-vehicle at 0.5 second
 - Notable Objects: car at (-3.70,13.08)
  Potential Effects: within the safe zone of the ego-vehicle at 0.5 second
*****Task Planning:*****
Driving plan: MOVE FORWARD WITH A DECELERATION

Figure 10:  An example of chain-of-thought reasoning and task planning. Cont'd.

**Autonomous Driving Planner**
Role: You're an autonomous vehicle's brain. Plan a 3-second safe trajectory to avoid obstacles.

Context:
- Coordinates: X-axis is perpendicular, and Y-axis is parallel to the direction you're facing. You're at point (0,0). Units: meters.
- Goal: Plan a 3-second route using 6 waypoints (0.5s intervals).

Inputs:
1. Ego States (important): Current stats (velocity, acceleration), past trajectory, goal direction.
2. Perception Results.
3. Past Experiences (important): Previous similar experiences with confidence scores and referenced planned trajectory.
4. Traffic Rules.
5. Reasoning (important): Notable objects affecting your plan and a top-level driving plan.

Task:
- Based on inputs, plan a safe, feasible 3-second trajectory of 6 waypoints.

Output:
Planned Trajectory:
[(x1,y1), (x2,y2), ... , (x6,y6)]

*****Environmental Information:*****
Current State:
 - Velocity (vx,vy): (-0.01,0.92)
 - Heading Angular Velocity (v_yaw): (0.00)
 - Acceleration (ax,ay): (-0.00,-0.50)
 - Can Bus: (-0.74,0.14)
 - Heading Speed: (0.95)
 - Steering: (-0.02)
Historical Trajectory (last 2 seconds): [(-0.07,-6.43), (-0.05,-4.34), (-0.02,-2.32), (-0.01,-0.91)]
Mission Goal: FORWARD

Front object detections:
Front object detected, object type: car, object id: 2, position: (4.36, 9.56), size: (1.86, 4.72)
Front object detected, object type: car, object id: 3, position: (-3.70, 13.08), size: (2.01, 4.92)

Future trajectories for specific objects:
Object type: car, object id: 2, future waypoint coordinates in 3s: [(4.36, 9.56), (4.36, 9.56), (4.36, 9.57), (4.36, 9.57), (4.36, 9.56), (4.36, 9.56)]
Object type: car, object id: 3, future waypoint coordinates in 3s: [(-2.66, 13.82), (-1.69, 14.79), (-0.99, 16.13), (-0.25, 17.73), (0.19, 19.42), (0.57, 21.35)]

Distance to both sides of road shoulders of selected locations:
Location (4.36, 9.56) distance to left shoulder is 16.5m and right shoulder is 0.5m
Location (-3.70, 13.08) distance to left shoulder is 13.0m and right shoulder is 8.5m

*****Past Driving Experience for Reference:*****
Most similar driving experience from memory with confidence score: 0.75:
The planned trajectory in this experience for your reference:
[(0.01,0.98), (0.01,1.87), (0.02,2.40), (0.02,2.64), (0.02,2.76), (0.02,2.86)]

*****Common Sense:*****
- Avoid collision with other objects.
- Always drive on drivable regions.
- Avoid driving on occupied regions.
- Pay attention to your ego-states and historical trajectory when planning.
- Maintain a safe distance from the objects in front of you.

*****Chain of Thoughts Reasoning:*****
Thoughts:
 - Notable Objects: car at (4.36,9.56), moving to (4.36,9.57) at 0.5 second
   Potential Effects: within the safe zone of the ego-vehicle at 0.5 second
 - Notable Objects: car at (-3.70,13.08), moving to (0.57,21.35) at 2.5 second
   Potential Effects: within the safe zone of the ego-vehicle at 2.5 second
*****Task Planning:*****
Driving plan: MOVE FORWARD WITH A DECELERATION

Planned Trajectory:
[(0.01,0.77), (0.01,1.43), (0.02,1.92), (0.02,2.27), (0.02,2.47), (0.02,2.57)]

Figure 11: An example of motion planning.

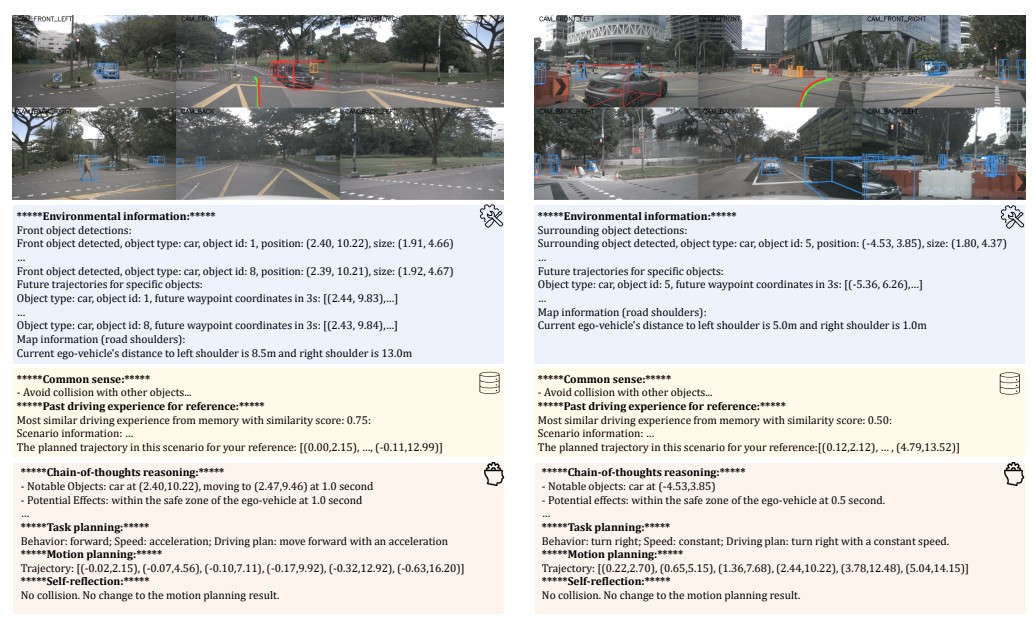

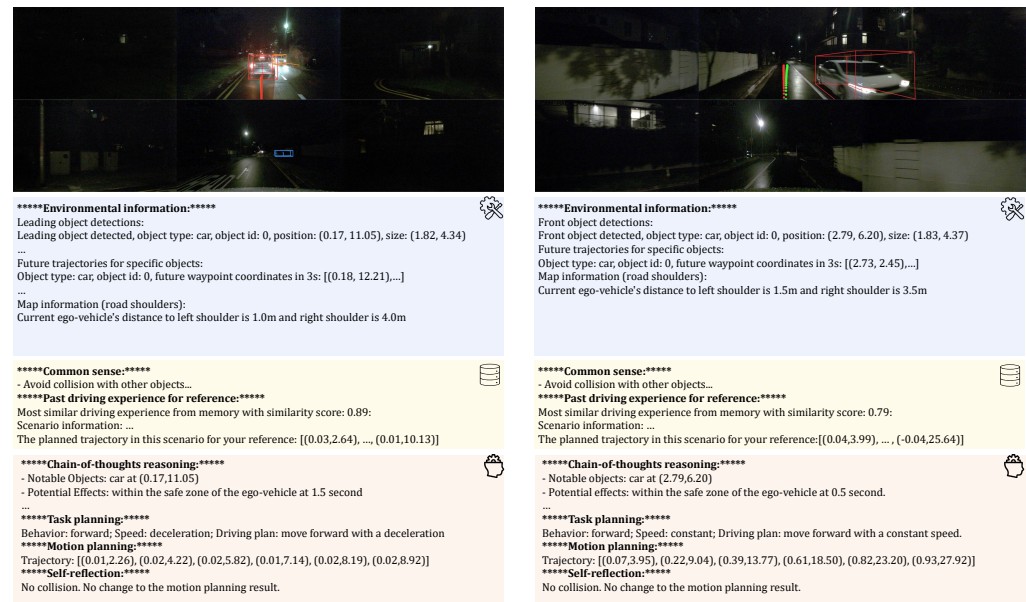

Figure 12: Interpretability of Agent-Driver.

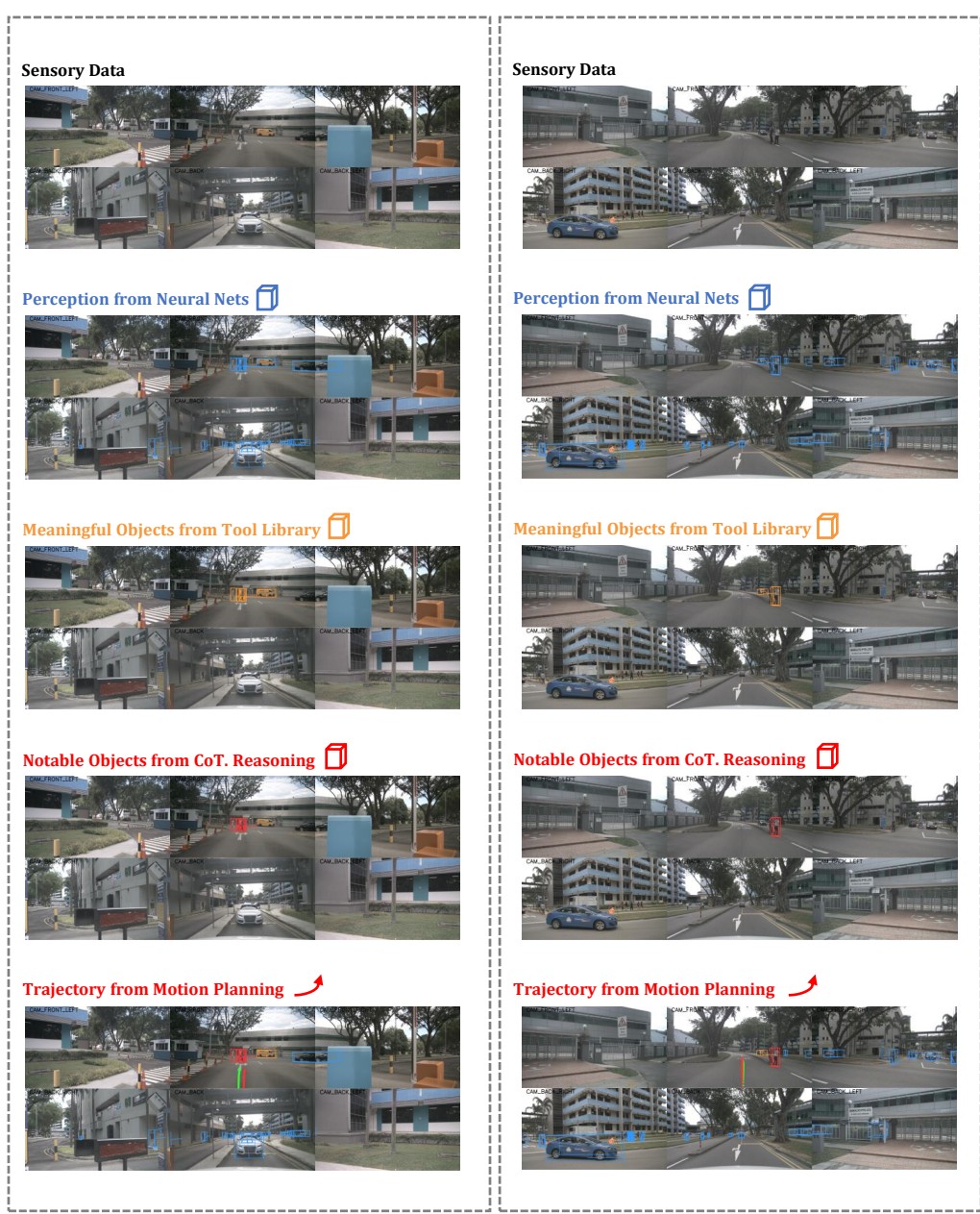

Figure 13: Visualization of how Agent-Driver progressively identifies critical objects and performs motion planning.

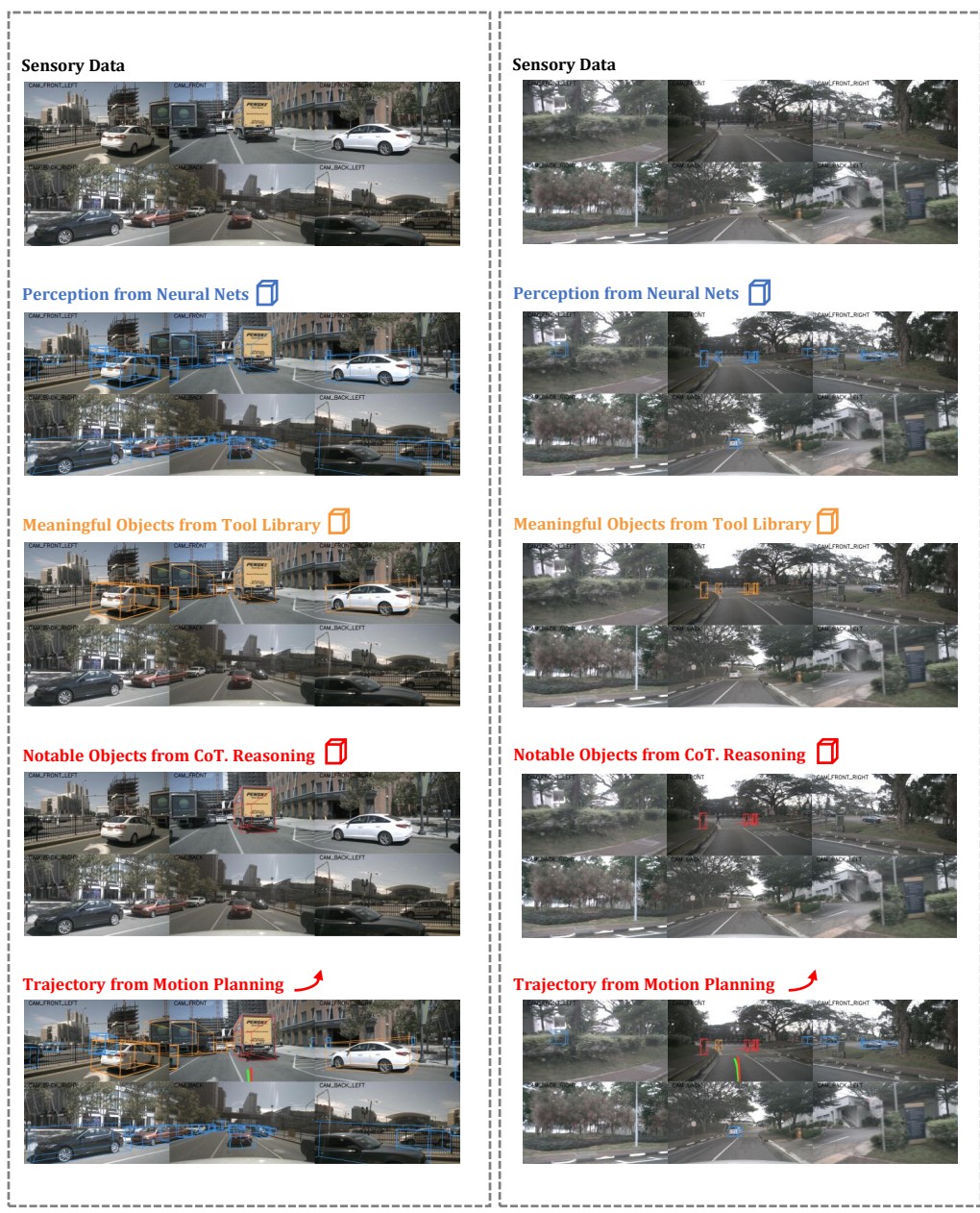

Figure 14: Visualization of how Agent-Driver progressively identifies critical objects and performs motion planning.

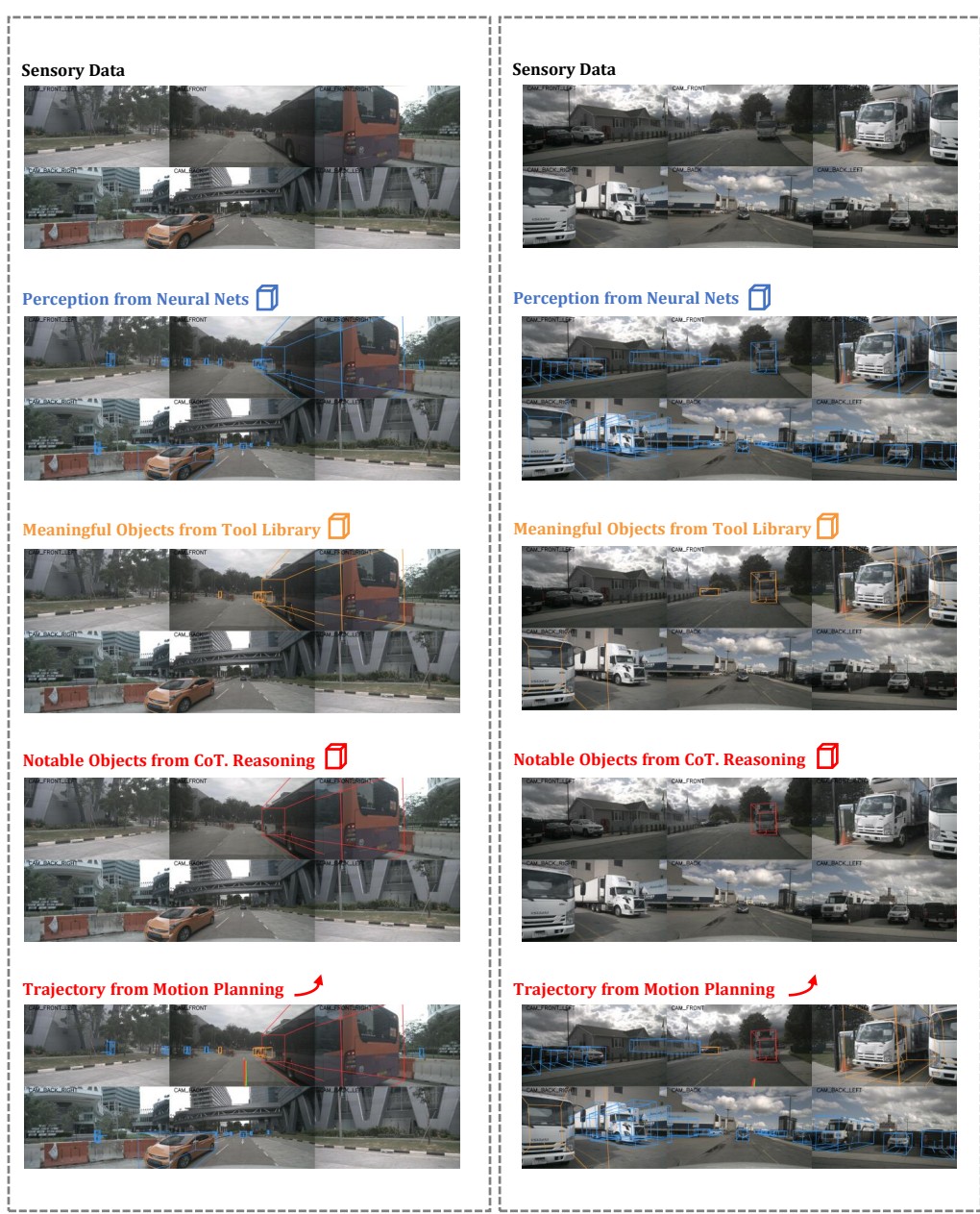

Figure 15: Visualization of how Agent-Driver progressively identifies critical objects and performs motion planning.

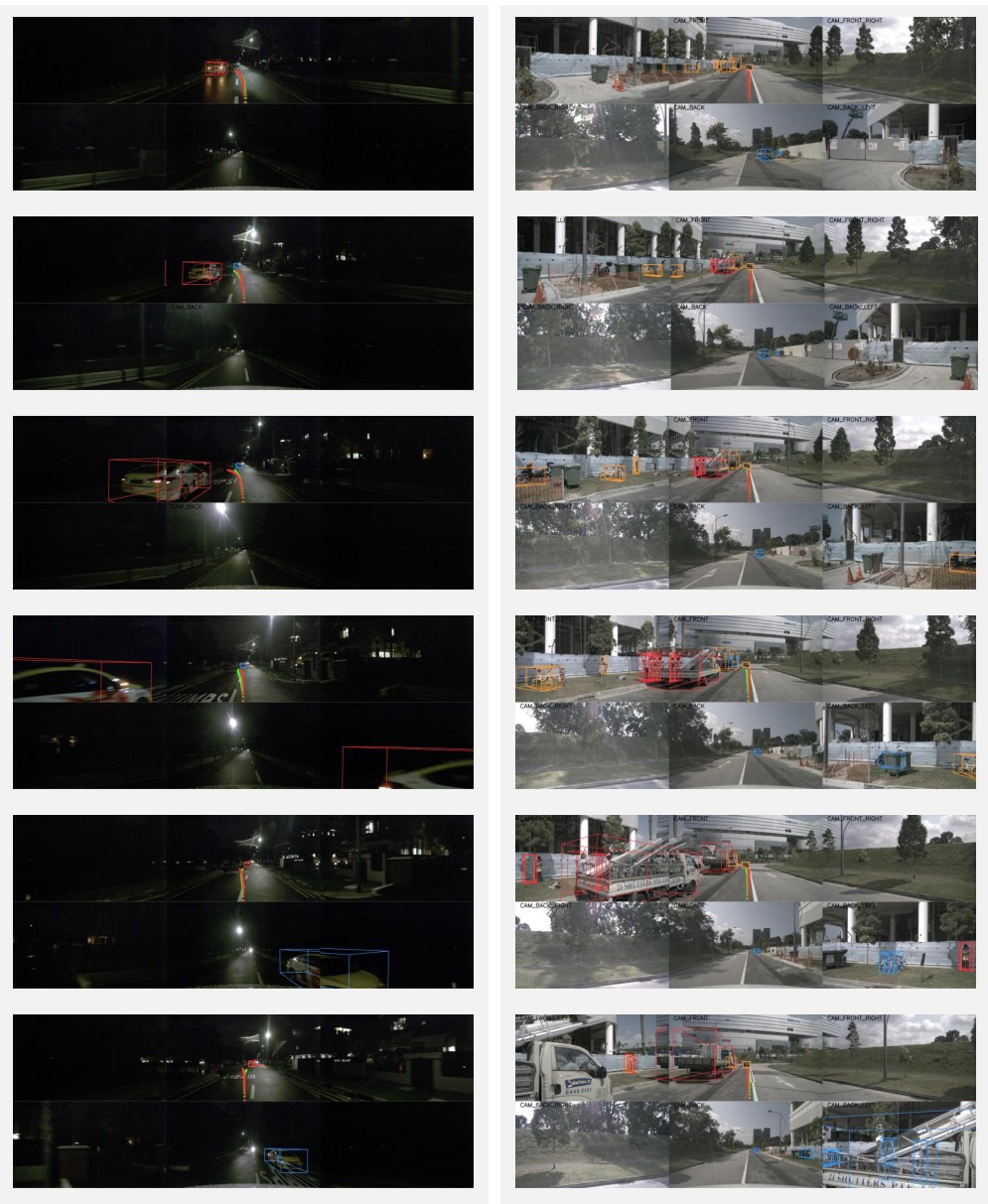

Figure 16: Driving video of Agent-Driver.

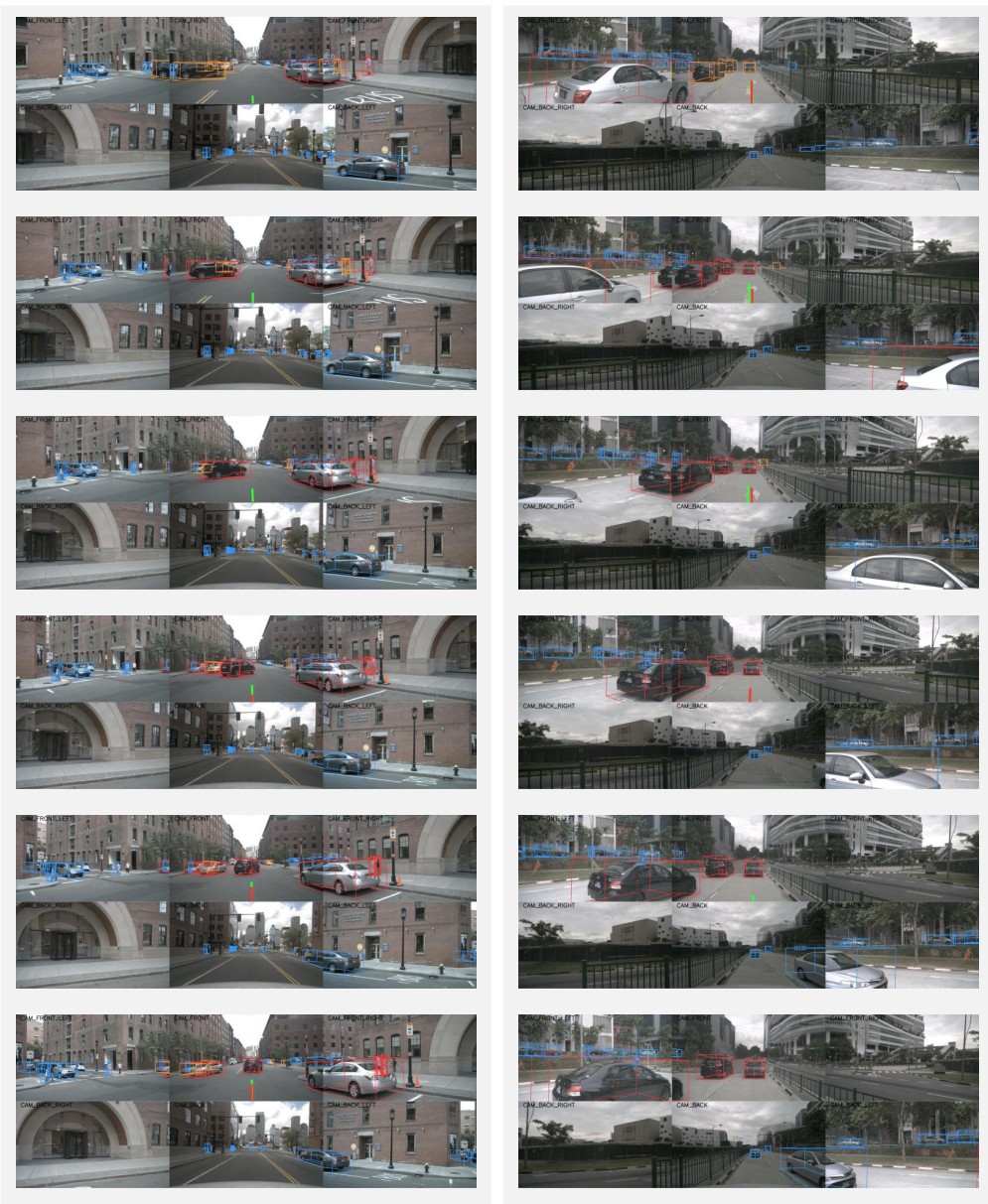

Figure 17: Driving video of Agent-Driver.

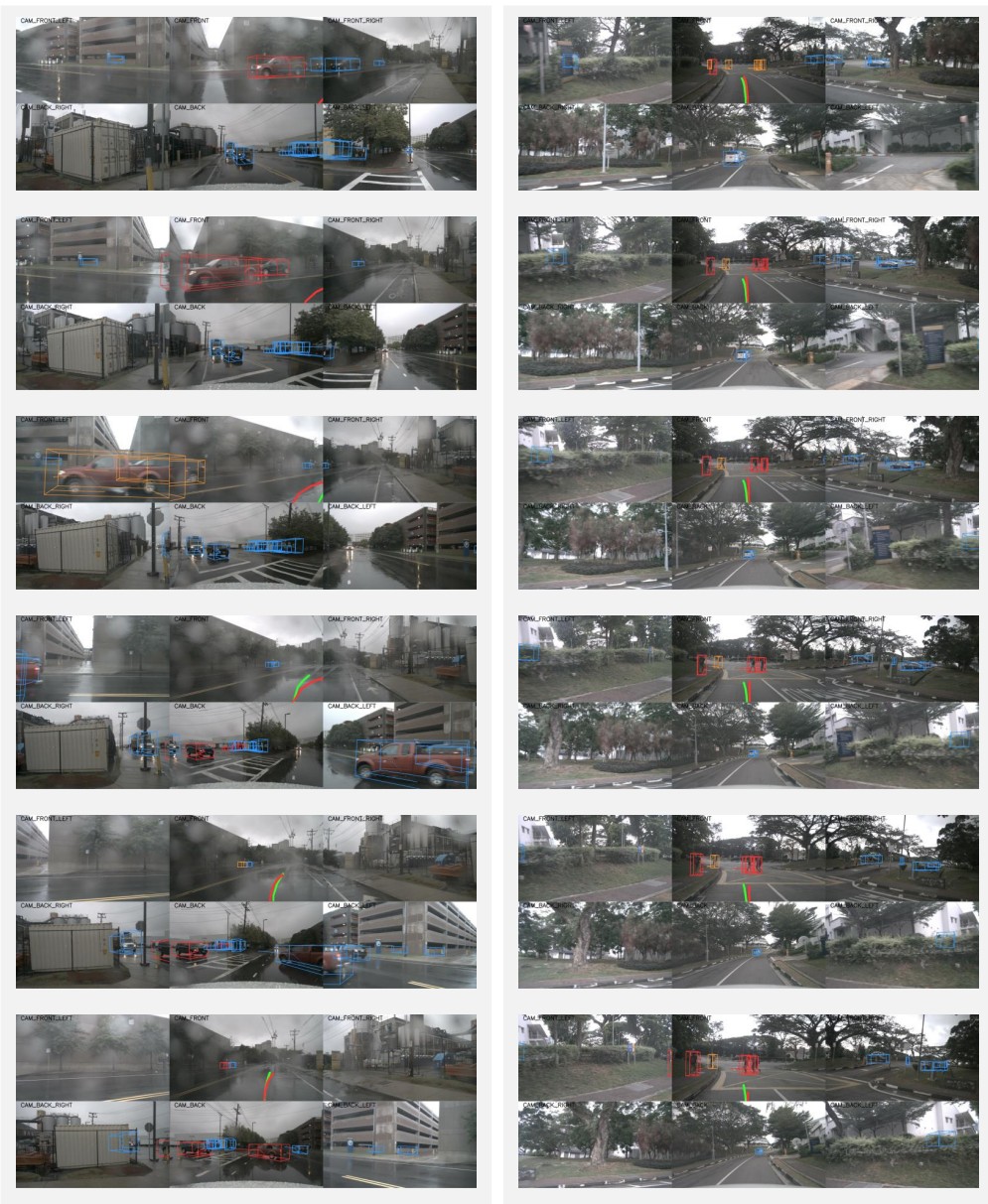

Figure 18: Driving video of Agent-Driver.

