# OpenReview forum: "A Language Agent for Autonomous Driving"
_colmweb.org/COLM/2024/Conference — COLM_

### Official Review · Reviewer_1NQx · 2024-05-01

**Rating:** 7
**Confidence:** 5
**Ethics Flag:** 1

**Summary:**

This paper introduces an LLM-supported agent named Agent-Driver, and the proposed system consists of a multifunctional tool library, a cognitive memory, and a reasoning engine. The cognitive memory contains commonsense memory and experience memory. The reasoning engine contains four parts: chain-of-thought reasoning, task planning, motion planning, and self-reflection. It conducted experiments on open-loop and close-loop driving challenges, and achieved SoTA results on the nuScenes benchmark.

**Questions To Authors:**

Have the authors considered choosing a loadable LLM model such as Vicuna instead of calling the LLM Api?

**Reasons To Accept:**

-	Agent-Drivers are redefining the traditional framework of autonomous driving by embedding human-like reasoning and decision-making capabilities. The system uniquely integrates a dynamic perception and prediction tool library and cognitive memory to achieve a more humane and anthropomorphic driving process.
-	This paper is well structured and highly readable.
-	It surpasses existing autonomous driving methods, showing over 30% collision improvements in motion planning, and excels in few-shot learning on the nuScenes benchmark.

**Reasons To Reject:**

-	Using LLMs as agents to achieve reasoning and memory functions is not a unique design of this work. Previous research, such as the DiLu [1] framework, has also implemented similar concepts.
[1] Wen L, Fu D, Li X, et al. Dilu: A knowledge-driven approach to autonomous driving with large language models. ICLR, 2024.
-	Although the paper proposes a novel paradigm for autonomous driving, the inherent latency issue due to reliance on API calls to LLM is still a concern. The author also mentioned this in the limitation, which is worthy of recognition. The delay issue is directly related to the real-time applicability of this method in actual driving scenarios.

---

> ### Author Rebuttal · Authors · 2024-05-27
>
> Thanks for your valuable comments!
>
> - Comparison with DiLu: Thanks for pointing this out! We have discussed the comparisons with DiLu in the related works (please refer to the last paragraph of appendix A). Our method is fundamentally different from DiLu in many aspects:
>
> (1) Architecture: Compared to DiLu, Agent-Driver has a more comprehensive architecture and has a better capability to handle realistic driving challenges. Our tool library includes a comprehensive perception and prediction toolkit which covers detection, prediction, occupancy, and mapping. Our memory is divided into a commonsense memory that stores driving knowledge and an experience memory that stores past driving experiences. Our reasoning engine is more structured and contains reasoning, task and motion planning, and self-reflection. All the above-mentioned components were not proposed or included in DiLu.
>
> (2) Experimental evaluation: Compared to DiLu which was evaluated in a virtual and simple  highway environment, Agent-Driver is evaluated on a real-world dataset nuScenes, and a more challenging close-loop simulator Carla. Evaluation in these realistic and challenging driving scenarios better reflects the effectiveness of our method.
>
> We will add more discussions of DiLu in the related works. Thanks for your comments!
>
> - Latency issue: In addition to the limitations we have discussed in the paper, we'd also like to point out that recently more and more papers [1-2] have been actively working on accelerating the inference of LLMs, which greatly reduces the latency of LLMs. In the meantime, due to hardware development, autonomous vehicles' onboard computational power also evolves rapidly. Given these two factors, we believe the inference restriction will be resolved in the near future. We will add more explanations to the limitation section. Thanks for your advice!
>
> - Loadable LLM model instead of API calls: We have provided experiments using Llama2-7b as a core LLM in Section 3.5 and Table 3. The results show that using a loadable LLM like Llama2 can lead to a decent performance.
>
> We hope these responses properly address your concerns. We remain fully committed to addressing any additional concerns that may arise. Thank you again for your valuable comments!
>
> [1] Liu et al. Deja Vu: Contextual Sparsity for Efficient LLMs at Inference Time.
> [2] Lin et al. Awq: Activation-aware weight quantization for llm compression and acceleration.

---

> > ### Comment · Reviewer_1NQx · 2024-06-05
> >
> > The rebuttal has addressed all my concerns and I am happy to raise my rating to 7, accept.

---

> > > ### Author Response · Authors · 2024-06-05
> > >
> > > Thank you for all these constructive comments to help improve this submission!

---

### Official Review · Reviewer_THFV · 2024-05-11

**Rating:** 7
**Confidence:** 4
**Ethics Flag:** 1

**Summary:**

This paper presents Agent-Driver, an LLM-based agent designed for autonomous driving. Agent-Driver operates other perceptual neural models as tools and is equipped with a reasoning engine that facilitates chain-of-thought reasoning, task planning, motion planning, and self-reflection. The model's promising performance is evidenced by both open-loop and closed-loop evaluations. Overall, this paper is well-written and features robust experimental work, with some minor concerns over the setups and reproducibility.

**Questions To Authors:**

Questions:
1. Are the same neural networks employed in the CARLA evaluation?

2. As an LLM agent, is Agent-Driver capable of giving justifications to actions, e.g., as defined in the BDD-X benchmark? It might be interesting (though not required) to compare with effective baselines on BDD-X (ADAPT, DriveGPT-4, DriVLMe, etc).

Suggestions:
1. I was confused by the terms “commonsense memory and experience memory”, I think in the memory modeling community these are close to semantic and episodic memories.

**Reasons To Accept:**

1. The Agent-Driver: The proposed Agent-Driver framework applies an LLM agent to the autonomous driving domain. This agent operates other perceptual neural models as tools and includes a sophisticated reasoning engine. The technical execution regarding the LLM agent is thoroughly solid.

2. Experiments: The experimentation is comprehensive. It includes both closed-loop and open-loop evaluations, alongside various ablation studies that test different LLM variants and neural module variants, providing a thorough examination of the system's capabilities.

**Reasons To Reject:**

1. Multimodal Domain Integration: Given that autonomous driving is a highly multimodal domain, this paper utilizes perception modules from prior work, such as UniAD. However, considering the availability of multimodal LLM agents like DriveGPT-4, the paper should justify why it opts to use other modality processors as tools instead of leveraging a unified multimodal approach.

2. Language-related Benchmarks: It would be beneficial for the experiments to be conducted using the BDD-X dataset, which focuses on the language capabilities and interpretability of driving agents. This would provide a clearer assessment of how well the Agent-Driver can handle real-world language and driving scenarios.

3. Clarity in Experimental Setup: The descriptions of the closed-loop experiments lack clarity. It is not specified whether the same neural networks are employed across different test setups, which is crucial for assessing the consistency and reliability of the results.

---

> ### Author Rebuttal · Authors · 2024-05-27
>
> Thanks for your valuable comments!
>
> - Multimodal Domain Integration: Thanks for your comments! While leveraging a unified multi-modal LLM to handle all driving challenges is a promising research direction, a multi-modal LLM that directly maps sensory inputs to actions generally lacks sufficient robustness, as it cannot fully utilize important intermediate representations like detections, predictions, and maps to avoid collision. On the other hand, we choose a fundamentally different approach, where we still leverage those modular perception modules as tools and introduce LLM as a scheduler and reasoning engine, which could benefit from both these accurate perception results and the reasoning and planning ability of LLM. Which approach is better remains an open question, and we're open to insightful discussions.
>
> - Language-related Benchmarks: Thanks for your suggestions! Compared to the multi-modal driving agents that focus more on language-based driving scene understanding or driving question-answering, our Agent-Driver has a different focus: we aim to address the planning and decision-making problems in real-world driving. Hence, we choose to evaluate our approach on the broadly used driving challenges like nuScenes and Carla. Language benchmarks, such as BDD-X for VLM driving agents, are out of the scope of the current paper. Also, our system does not have a VLM available for BDD-X evaluation. However, we agree that scene description and understanding are very important to autonomous driving, and we plan to integrate a multi-modal LLM as a tool for scene understanding in a follow-up work. Thanks for your suggestion.
>
> - Environmental setup: Thanks for your suggestions! For closed-loop experiments on Carla, we leveraged the perception modules in LAV [1] and kept the other parts of our system the same. We also followed the same training setting and evaluation protocols in [1] for a fair comparison. We will add more details about the closed-loop experiments to our paper. Thanks for your suggestions!
>
> - Commonsense memory and experience memory. Sorry for the confusion. We will change the terminology to semantic and episodic memories and provide further explanations in our paper. Thanks for pointing this out!
>
> We hope these responses properly address your concerns. Feel free to let us know if you have further concerns and we would be thrilled to address them. Thank you again for your valuable comments!
>
> [1] Chen et al. Learning from All Vehicles.

---

> > ### Comment · Reviewer_THFV · 2024-05-31
> > **Comments from Reviewer**
> >
> > Thanks for the reply. I would like to keep my positive evaluation of this work.

---

> > > ### Author Response · Authors · 2024-06-05
> > >
> > > Thank you again for your constructive comments and suggestions. If we have successfully addressed your questions, we would strongly appreciate an increased score! Otherwise, feel free to let us know and we are happy to provide additional experiments and/or discussion to allay your concerns.

---

### Official Review · Reviewer_zg5k · 2024-05-13

**Rating:** 8
**Confidence:** 4
**Ethics Flag:** 1

**Summary:**

The paper presents Agent-Driver, a system utilizing an LLM as a driving agent. The system consists of several modules: tool library for detection, prediction, occupancy, and map modules, cognitive memory which allows search in past driving scenarios, reasoning engine which prompts an LLM for chain-of-thought reasoning, task planning, motion planning and self-reflection (which are all steps of giving prompts in a specified format).
The system is evaluated with experiments on open-loop driivng on the nuScenes dataset and closed-loop driving in the CARLA simulator.
The system is compared to several other recent works with LLMs for autonomous driving. The results show improvement in route completion for closed-loop driving and improvement on most metrics for open-loop driving.

**Reasons To Accept:**

The presented system builds on top of recent line of research that uses LLMs for autonomous driving. In contrast to some recent work, the system uses components which specialize indifferent aspects of the driving task, and according to the reported results, this setup seems to be beneficial.
The components are described in a good detail and the architecture of the system is mostly understandable.

**Reasons To Reject:**

Not really a reason to reject, but maybe something that should be explained. The text prompts to the LLM in the reasoning engine resemble the one in GPT-Driver. Maybe this could be commented. What is the difference between the reasoning engine and the implementation in GPT-Driver? Is the current paper an extension that adds memory and tool library (this does not mean this is bad, but just that maybe it could be explained for clarity)?
In the current paper, the prompt is given step-by-step, how is this information passed to the LLM? As separate messages? Does it make any difference in comparison to having one big message, or does this separation make it easier for the model to learn?

---

> ### Author Rebuttal · Authors · 2024-05-26
>
> Thanks for your valuable comments!
>
> - What is the difference between the reasoning engine and the implementation in GPT-Driver? There are several key differences between our reasoning engine and the one in GPT-Driver:
>
> (1) We decoupled the chain-of-thought reasoning, task planning, and motion planning as separate steps to the LLM. Compared to generating both reasoning and planning in a single pass as in GPT-Driver, we empirically found that such separation eases the learning and prediction process, and yields a better result.
>
> (2) We use in-context learning for instructing the LLM to perform reasoning and task planning, which encourages better diversity and generalization ability. While in GPT-Driver, it uses fine-tuning.
>
> (3) We additionally introduce a self-reflection stage to further rectify the motion planning results, which leads to fewer collisions and better safety.
>
> - Information flow in the current method: We employed multiple rounds of conversations and used separate messages at each stage to the LLM. Compared to sending all messages as a whole to the LLM, we empirically found that separating the messages makes the LLM focus on one task at each stage, which eases the learning process and yields a better result.
>
> We hope these responses properly address your concerns. We are committed to addressing any further concerns. Thank you again for your valuable comments!

---

> > ### Comment · Reviewer_zg5k · 2024-06-05
> >
> > Thank you for the response! I am keeping my score for the paper.

---

> > > ### Author Response · Authors · 2024-06-05
> > >
> > > Thank you again for the positive evaluation and constructive comments!

---

### Official Review · Reviewer_sSRF · 2024-05-21

**Rating:** 9
**Confidence:** 4
**Ethics Flag:** 1

**Summary:**

The paper presents a novel framework using LLMs for autonomous driving planning tasks. The proposed method integrates 1) a tool library for perception and prediction, 2) a memory module of commonsense and experience, and 3) LLM-based reasoning module that performs chain-of-thoughts reasoning, and motion planning. The paper conducts comprehensive experiments to demonstrate 1) the effectiveness of the methods in terms of L2 distance to ground truth and collision rates, 2) few shot learning ability by training on a small percentage of data, and 3) performance difference for in-context learning and fine-tuning. The paper also includes comprehensive details of the method and implementation in the appendix (20+ pages), which is laudable. The paper is well written, and easy to follow. The diagrams are illustrative. Experiments are reasonable.

Overall, it's a forward-outlook novel paper that showcases the effectiveness of LLMs applied to autonomous driving. This work is a valuable addition to the field. The results are worth sharing. I hesitate between top 5% and top 15% accept papers, as this paper is already pretty well-known in the field of autonomous driving, and is considered one of the early notable papers that apply end-to-end LLM agents to autonomous driving.

**Reasons To Accept:**

- The paper presents a novel framework using LLMs for autonomous driving planning tasks. The proposed method integrates 1) a tool library for perception and prediction, 2) a memory module of commonsense and experience, and 3) LLM-based reasoning module that performs chain-of-thoughts reasoning, and motion planning.
- The paper conducts comprehensive experiments to demonstrate 1) the effectiveness of the methods in terms of L2 distance to ground truth and collision rates, 2) few shot learning ability by training on a small percentage of data, and 3) performance difference for in-context learning and fine-tuning.
- The paper also includes comprehensive details of the method and implementation in the appendix.
- The paper is well written, and easy to follow. The diagrams are illustrative.

**Reasons To Reject:**

Some minor comments and questions (not necessarily reasons to reject the paper):

I would be interested in learning
- Training and inference costs. The paper only mentions it in Appendix G as "unable to obtain the inference time". It'd be helpful to provide a rough estimate of at least the magnitude of inference time compared to non-LLM based state-of-the-art-ish baselines.
- Where does the method struggle? The paper provides a couple of failure examples but does not provide a description on where the LLM-based driver likely to fail. The limitation paragraph in the appendix only mentions inference time.
- How much does CoT help in long-tail / challenging driving scenarios? The paper does not demonstrate it despite the opening example in the Intro section.

Experiments
- The paper reports L2 distance to ground truth and the collision metrics. Showing additional metrics such as progress, and comfort would strength the results.
- What does "invalid" mean in sec3.6?
- The main text does not specify the "in-context learning" method. From the appendix, in-context learning learns from "four human-annotated examples". Does fine-tuning also use such small number of samples? Would be better to clarify in either the main text or appendix to establish fair comparisons.

---

> ### Author Rebuttal · Authors · 2024-05-26
>
> Thanks for your valuable suggestions!
>
> - Costs: inference time using API calls takes 500 ms on average, which also includes the network communication latency. We note that many papers are actively working on accelerating LLM inference, and vehicles' onboard computational power also evolves rapidly. Therefore, we believe this limitation will be resolved soon. We will add more explanations to the limitation section.
>
> - Where does the method struggle: similar to other driving methods, our method is limited by the perception results from the tool library. If the tool library provides noisy results, the reasoning engine might not make appropriate decisions. Figure 2 in Appendix F.3 better illustrates this issue. Future work to address this limitation includes leveraging multiple perception models to provide increased robustness.  We will add more explanations to this section.
>
> - How much does CoT help in long-tail / challenging driving scenarios: we provided Figure 1 in Appendix F.1 as a qualitative study showing the effects of CoT in challenging scenarios. The figure shows that when driving near a construction area, CoT helps generate more suitable driving trajectories that effectively avoid approaching this construction area.
>
> - Progress and comfort as evaluation metrics: in addition to the L2 and collision metrics, we have also included Route Completion and Driving Scores as metrics for driving progress and comfort in Carla experiment (please refer to Table 1 and section 3.2 of the main paper). The Route Completion denotes the progress of driving on a route, and the Driving Score additionally takes comfort and safety into calculation.
>
> - What does "invalid" mean in sec3.6: as the reasoning engine needs to generate waypoint coordinates for motion planning, we denote "invalid" as the waypoint outputs contain non-numerical values. We note that fine-tuning enables robust alignments and our reasoning engine can always generate valid waypoints with a few fine-tuning data. We will add more explanations to this section.
>
> - In-context learning: we will add more clarifications in the main paper regarding the concept of in-context learning and fine-tuning. For fine-tuning, we can use more samples, as it is not restricted by the context window like in-context learning.
>
> We hope these responses properly address your concerns. Thank you again for your valuable suggestions! We are committed to providing additional responses should you have any further comments.

---

> > ### Author Response · Authors · 2024-06-06
> >
> > Thank you again for such positive evaluation and constructive comments! We will greatly appreciate an increased score if we have successfully answered your questions :) Otherwise, we would love to provide additional responses to address any further concerns.

---

### Decision · Program_Chairs · 2024-07-10

**Decision:**

Accept

**Comment:**

A well-done application paper that applies end-to-end LLM agents to autonomous driving, which all the reviewers unanimously liked. The paper is well-written and easy to follow, with clear diagrams. The technical execution regarding the LLM agent is solid, and the experimentation is comprehensive. Agent-Driver's unique integration of a dynamic tool library, cognitive memory, and a sophisticated reasoning engine distinguishes it from prior works. This would be a great addition to the conference.

Pros:
- Novel framework integrating LLM with traditional autonomous driving components.
- Strong performance on open-loop and closed-loop driving challenges including the challenging nuScenes benchmark

Cons:
- Lack of evaluation on language-related benchmarks like BDD-X.
- Need for clearer explanation of the differences between Agent-Driver's reasoning engine and other implementations (e.g., GPT-Driver).